# Sensitivity analysis enlightens effects of connectivity in a Neural Mass Model under Control-Target mode

Anaïs Vallet[1], Stéphane Blanco[2], Coline Chevallier[2,3], Francis Eustache[1], Jacques Gautrais[2,3]*, Jean-Yves Grandpeix[4], Jean-Louis Joly[2], Shailendra Segobin[1], Pierre Gagnepain[1]

1 Normandie Univ, UNICAEN, PSL Research University, EPHE, INSERM, U1077, CHU de Caen, GIP Cyceron, Neuropsychologie et Imagerie de la Mémoire Humaine, Université de Normandie, Caen, France, 2 CNRS, INPT, UPS, LAPLACE, Université de Toulouse, Toulouse, France, 3 Centre de Recherches sur la Cognition Animale (CRCA), Centre de Biologie Intégrative (CBI), CNRS, UPS, Université de Toulouse, Toulouse, France, 4 LMD/IPSL, CNRS, École Polytechnique, ENS, Sorbonne Université, Paris, France

* jacques.gautrais@cnrs.fr

## Abstract

*Biophysical models* of human brain represent the latter as a graph of inter-connected neural regions. Building from the model by Naskar et al. (Network Neuroscience, 2021), our motivation was to understand how these brain regions can be connected at neural level to implement some inhibitory control, which calls for inhibitory connectivity rarely considered in such models. In this model, regions are made of inter-connected excitatory and inhibitory pools of neurons, but are long-range connected only via excitatory pools (mutual excitation). We thus extend this model by generalizing connectivity, and we analyse how connectivity affects the behaviour of this model. Focusing on the simplest paradigm made of a Control area and a Target area, we explore four typical kinds of connectivity: mutual excitation, Target inhibition by Control, Control inhibition by Target, and mutual inhibition. For this, we build an analytical sensitivity framework, nesting up sensitivities of isolated pools, of isolated regions, and of the full system. We show that inhibitory control can emerge only in Target inhibition by Control and mutual inhibition connectivities. We next offer an analysis of how the model sensitivities depends on connectivity structure, depending on a parameter controling the strength of the self-inhibition within Target region. Finally, we illustrate the effect of connectivity structure upon control effectivity in response to an external forcing in the Control area. Beyond the case explored here, our methodology to build analytical sensitivities by nesting up levels (pool, region, system) lays the groundwork for expressing nested sensitivities for more complex network configurations, either for this model or any other one.

**Data availability statement:** All relevant data are within the manuscript and its Supporting Information files.

**Funding:** This study was supported by the French National Research Agency under France 2030, ANR-10-EQPX-0021 Programme 13-Novembre to FE and AV's PhD was funded by Institut National de la Santé et de la Recherche Médicale and Normandie Region. The funders had no role in study design, data collection and analysis, decision to publish, or preparation of the manuscript.

**Competing interests:** The authors have declared that no competing interests exist.

## Author summary

Biophysical models of the human brain involve representing how its neurons interact among each other. In the brain, connectivity can be excitatory or inhibitory but only excitation is usually considered in biophysical models. Here, we propose a biophysical model including inhibition to give account of inhibitory control. We consider the simplest model that consists of a Control and a Target area, and we explore four types of connectivities: mutual excitation, Target inhibition by Control, Control inhibition by Target and mutual inhibition. We develop an analytical expression of the system's response and show that inhibitory control happens only in cases of Target inhibition by Control and mutual inhibition. We illustrate how the system responds when the control area is excited by an external stimulus, and highlight the role of a parameter that drives the strength of self-inhibition within the Target region. Our analytical framework paves the way to study more complex brain network configurations using such biophysical models.

# 1 Introduction

## 1.1 Motivation

In recent years, a large effort has been made to design and implement large-scale simulations of human brain, in relation to functional neuroimaging data (EEG, MEG, functional MRI, ...) in order to draw inferences regarding neurophysiological mechanisms [1–5]. In this domain, the class of so-called biophysical models represent the brain as a graph of inter-connected neural masses, and the neural dynamics are used to link structural connectivity (aka anatomical connectivity, or structural connectome) to functional imaging data (e.g., BOLD signals from fMRI) [6–11]. A widespread practice is to use recordings of fMRI to infer the so-called functional connectivity from correlations between time courses of areas' activity level. This functional connectivity can then be instrumental to tune biophysical models' parameters [11]. In such approaches, structural connectivity among cortical areas plays then a crucial role into shaping dynamics and a huge effort has been made to obtain and characterize a precise mapping of brain connectivity by *in vivo* tractography [12–16].

Structural connectivity among cortical areas is known to be supported by long-range connections that can be only excitatory, and in most biophysical models where neural masses activity level are represented by one state variable, they are considered as mutually excitatory. There are however some contexts in which the influence from one area upon another one is clearly considered as inhibitory by nature, the most prominent paradigm being the tasks involving inhibitory control [17–20]. Inhibitory control is a mechanism depending on prefrontal executive functions, that enables the brain to override or cancel reflexive actions, memories, or emotions by deactivating their associated representations or processes [19].

Functional MRI (fMRI) reveals that inhibitory control is characterized by a decrease in BOLD activity in some Target brain regions when some Control brain regions are subject to an increase in BOLD activity. Analyses of effective connectivity further reveals that the downregulation of the Target regions is mediated by a negative top-down coupling originating from Control regions (e.g., [21]).

In such context, it must be considered that the activity coming from a control region can have an inhibitory effect upon target populations, namely inhibitory control should translate increasing activities in some control areas into decreased activities in their target areas, an effect that is currently lacking in recent dynamic mean field (DMF) model (e.g., [1,22]).

These models should therefore be extended to include long-range connectivity reflecting the feedforward activation of inhibitory interneurons via polysynaptic pathways, and drive suppression of activity in Target region.

As a consequence, there is a clear need to analyse and understand the effect of generalizing signs of connectivity upon the behaviour of such systems. In this paper, we offer an analytical account of how changing the signs of connectivity can affect activity levels in a widely used biophysical model [7]. We consider the simplest configuration with two areas connected with four scenarios of connectivity, within a control area - target area paradigm.

We conduct this analysis using sensitivities analysis, namely how a chosen observable (e.g., the level of activity in target area) depends upon the set of parameters, once a connectivity scenario has been fixed.

For neural mass models, various approaches aimed at determining parameter dependence have been proposed in recent years [23–25]. This task is complicated by the large number of parameters involved (typically dozens).

One way to highlight the relative contribution of each parameter is to sample the entire parameter space, perform discrete time numerical simulations, and select certain observables on the numerical trajectory of the system over time. Then, these observables can be classified with regards to a chosen metrics (e.g., oscillation frequency), using methods like random forest [24], or swarm optimization [25] or bayesian inference [23].

By contrast, here we consider the term "sensitivity" in the strict sense of a functional partial derivative of the value of a state variable with respect to a parameter, and we give the corresponding analytical expression, upstream of any numerical resolution. To this end, we develop a methodological framework for constructing a hierarchy of nested sensitivities and analyze the structure of the sensitivities obtained from the target area responses, based on connectivity.

We present our full analysis, in the hope that our results may enlighten modelling choices for connectivity, in the community dealing with biophysical models of human brain.

## 1.2 The model

### 1.2.1 Modeling background.
We start from the dynamic mean field (DMF) model of excitatory/inhibitory neural populations first introduced by Deco et al. [7] (after [26]), in an effort to account for resting-state networks (RSNs). For each area, mean-field description summarises the corresponding underlying spiking neural network, in which excitatory pyramidal neurones and GABAergic inhibitory neurones are mutually interconnected. At the mean field scale of description, areas are interconnected by long range excitatory signals (NMDA-mediated connections), weighted by anatomical connectivity and a global coupling strength.

These mean field equations are then integrated numerically, spanning initial conditions, in order to study the stationary state landscape. They tuned the global inter-areal coupling strength in the model to fMRI-based functional connectivity in humans and showed that resting brain operates at the edge of multi-stability [7]. This DMF model was then used to explore more exhaustively the dynamics of the system [27], especially how noise propagation reflects the double effect of anatomical connectivity and slow dynamics around the spontaneous low-firing stationary state, resulting in the functional connectivity (i.e., pairwise correlations between cortical areas). Here, they show that global inter-areal coupling strength fitted to fMRI data rather indicates that brain operates at full multi-stability, yet with resting state being a stable stationary state. Furthermore, they show that pairwise correlations depend on the sensitivities of each area to the inputs coming

from other areas. Since these sensitivities depend in turn upon the activity levels at the considered stationary state, affecting these activity levels (e.g., by over-activating a subset of areas) results in a different stationary state, and hence to a different functional connectivity, while anatomical connectivity remains the same [27,28]. The model has then been used to infer anatomical connectivity from functional connectivity, relating structure and function [29], to explore the possibility of dynamical transitions between multiple RSNs depending on noise level [9], to suggest that human brain anatomical connectivity may be evolutionary tuned to display the largest diversity in activities of area clusters in response to an increased inter-regional coupling [10].

In a refinement of this model, Deco et al. [8] propose to better describe neural mass areas by explicitly considering that they are composed of two pools of neurones: an excitatory one and an inhibitory one, so that the local feedback inhibition can be locally constrained. Under these constraints, the empirical finding that intra-areal correlations should remain poor is recovered, whereas it is violated when using inter-areal coupling in single-pool modelling. With this choice, the intra-areal activity can be made decorrelated while the inter-areal correlations keep reflecting functional connectivity. Moreover, and as a consequence of this modelling choice, long range excitatory signals can now target either excitatory pools in other areas or their inhibitory pools, or both. We then turned to this version of the model since it allows to represent feedforward inhibition, when one area can stimulate the inhibitory pool of another area. Also, Deco et al. [8] consider the effect of stimulating some random subset (10%) of areas by exogenous stimuli, representing task-related signals. Interestingly, in this version, the dynamics yield only one stable stationary state at resting (for reasonable coupling strength avoiding epileptical divergence), and must be stimulated by these exogenous stimuli to switch state. They advocate that this refinement yield more robust prediction of functional connectivity, and more realistic responses to external stimuli [8]. This two-pool area model at resting state has been used by Glomb et al. to recover the spatio-temporal structure of RSNs from fluctuations around the single stationary state [30]. As a final extension, Naskar et al. justify the kinetic parameters driving the average synaptic gating variable based on GABA and glutamate concentrations [1]. We start from this version.

**1.2.2 Formal expression of the model.** In this model, the state variables at neural mass scale, $sn(t)$ and $sg(t)$, are called *average synaptic gating variables* and represent the open fraction of NMDA (resp. GABA) synaptic channels for the excitatory (resp. inhibitory) pools. For each area $i$, their dynamics are given by the following ODEs:

$$\begin{cases} \dot{sn}_i(t) = -\beta^E sn_i(t) & +\alpha^E T_{glu} rn_i(t)(1 - sn_i(t)) + \sigma\nu_i(t) \\ \dot{sg}_i(t) = -\beta^I sg_i(t) & +\alpha^I T_{gaba} rg_i(t)(1 - sg_i(t)) + \sigma\nu_i(t) \end{cases}$$

(1)

where the intermediate variables $rn(t)$ and $rg(t)$ represent the mean-field firing rates of each pool, which are given by input-output functions (inspired from [31]):

$$\begin{cases} rn_i(t) = \dfrac{a_E xn_i(t) - b_E}{1 - e^{-d_E(a_E xn_i(t) - b_E)}} \\ rg_i(t) = \dfrac{a_I xg_i(t) - b_I}{1 - e^{-d_I(a_I xg_i(t) - b_I)}} \end{cases}$$

(2)

and depend, in turn, upon the input currents $xn(t)$ and $xg(t)$, that sum up all incoming contributions:

$$\begin{cases} xn_i(t) = W_E I_0 & + W_+ J_{nmda} sn_i(t) & - J_{gaba_i} sg_i(t) & +GJ_{nmda}\sum_{j\neq i} C_{ij} sn_j(t) \\ xg_i(t) = W_I I_0 & + J_{nmda} sn_i(t) & - sg_i(t) \end{cases}$$

(3)

In these representations, coupling are made explicit in Eq 3, namely coupling are modelled as flows of information about fractions of open channels in the excitatory pools of other areas.

At the area scale, there are four couplings: self-excitatory coupling of the excitatory pool, excitatory coupling from the excitatory pool upon the inhibitory pool, self-inhibitory coupling of the inhibitory pool and inhibitory coupling from the inhibitory pool upon the excitatory pool (see Fig 1).

Coupling among areas are represented through the sum term where $C_{ij}$ represent normalised anatomical connectivity, $J_{nmda}$ represents excitatory synaptic coupling strength, and $G$ is a positive free parameter which scales the global coupling strength between areas. The parameter $J_{gaba_i}$ represents inhibitory synaptic coupling strength, and can be locally adjusted to regulate the balance between long-range excitation and local feedback inhibition so as to ensure homeostasis [8].

Note that without the so-called *basic input current*, $I_0$, the system activity would tend to zero and only noise $\sigma$ would remain, and it is set to a negligible level throughout the papers considered above.

**1.2.3 Generalizing connectivity.** In the model above, coupling between areas is only between excitatory pools. To allow flexibility, we introduce a further parameter $k_{E_{ij}} \in [0; 1]$ which modulates the fraction of long range excitatory input coming from area $j$ that will projects onto the excitatory pool of area $i$, the remaining part being considered to project onto its inhibitory pool. We also denote $B_{E_i}$ and $B_{I_i}$ the effective external input current so that they can vary among areas, and we introduce a parameter $J_-$ to get an homogenous expression.

With these extensions, our system only modifies Eq 3, which now reads:

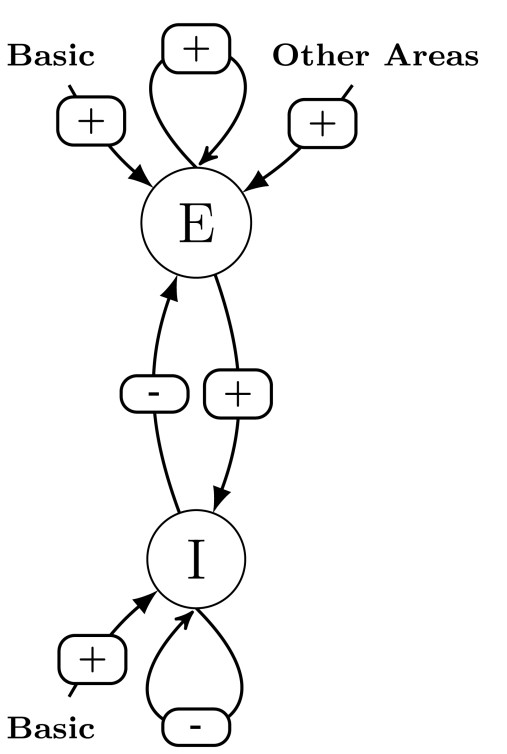

**Fig 1. Graphical representation of Eqs. 1 to 3 for a given area.**

$$\begin{cases} xn_i(t) = B_{E_i} \quad + W_+ J_{nmda} sn_i(t) \quad - J_{gaba_i} sg_i(t) \quad + \sum_{j \neq i} k_{E_{ij}} \kappa_{ij} sn_j(t) \\ xg_i(t) = B_{I_i} \quad + J_{nmda} sn_i(t) \quad - J_{-} sg_i(t) \quad + \sum_{j \neq i} (1 - k_{E_{ij}}) \kappa_{ij} sn_j(t) \end{cases}$$

(4)

where $\kappa_{ij} \equiv G \, J_{nmda} C_{ij}$ and $J_{-}$ = 1 nA in [1].

### 1.3 Analysis of connectivity effect based on sensitivities

**1.3.1 Control-Target mode.** Since we are interested in inhibitory control process, we can distinguish two kinds of areas, splitting the system into a subset of control areas, and a subset of target areas.

In the aim to analyse how connectivity among areas will shape the behavioral response of the system, the main picture is then to understand how target areas respond to upregulation applied to control areas, once we take into account that target areas can themselves project back to control areas and have a feedback effect upon their upregulation.

For a single area, a typical time course of the response of the excitatory pool to inputs is shown in Fig 2 (purple line) together with the corresponding BOLD signal (green line), as modelled in [33] (Numerical illustrations of the formal developments are all given below using the parameters values of reference listed in S1 File).

In response to alternate switching of $B_E$ value (between 0.382 and 0.482 nA), one can see that the activity level $sn(t)$ adjusts very quickly, whereas BOLD signal follows with an observable delay. This time scale separation holds in the control-target system (see section 3.4.1). At the neural mass level, we can then study the behavior of the system considering only the value $sn(t)$ at steady states, which are fixed points of the dynamics (with our range of parameters, all fixed points considered in the present paper are stable, see S2 File). Hence, we can address the question of the effect of connectivity looking only how the fixed points are affected by parameters, namely to predict how the target activity level will be affected.

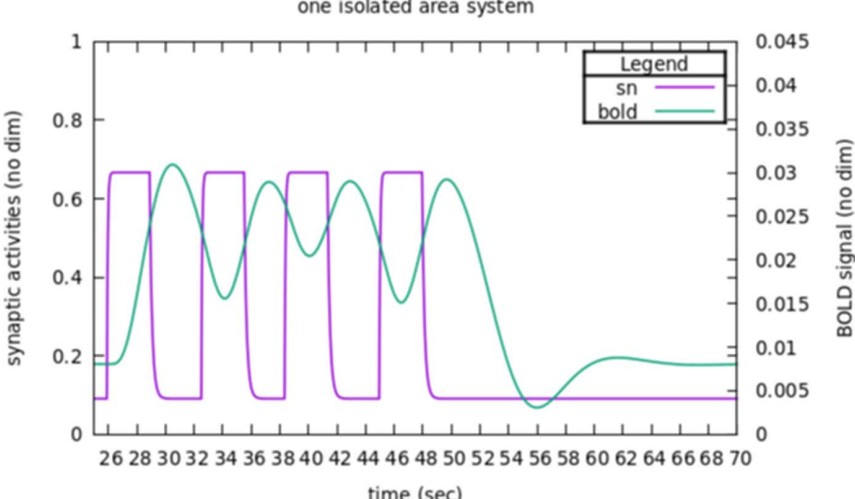

**Fig 2. Basic response of an isolated area to inputs.** The fraction of open channels *sn* and the corresponding BOLD signal are reported in reaction to successive stimulations. Time course of stimulations are made typical of a TNT task with stimulation steps during 3 s, separated by random intervals between 2.4 and 3.6 s (as in [32]). The BOLD signal is driven by *sn* as modelled in [33].

For this, we will develop a *sensitivity* analysis, in which we build an analytical expression for how the system responds to an infinitesimal perturbation upon some control areas, and how it affects activity levels in all areas (including those upon wich the perturbation is applied). The point of interest is then how the activity levels of target areas are modified, whether they are up-regulated or down-regulated compared to the baseline, which is given by the sign of sensitivities (i.e., the sign of the derivative of activity level with regards to the perturbation). Once such sensitivities expressions are established, they are used to analyze the behavior of the system, depending on the kind of connectivity.

**1.3.2 Formalizing the sensitivity of interest as a hierarchy of nested sensitivities.** The sensitivities are to be estimated at the fixed point corresponding to the resting state. Following Naskar et al. [1], the parameters $J_{gaba_i}$ will be adjusted on a per-area basis such that resting state corresponds to a firing rate of the excitatory pools prescribed at $rn_i^* = 3$ Hz.

Once this fixed point is established, we apply an infinitesimal perturbation $\delta B_{E_i}$ upon $B_{E_i}$ in control areas, and we will observe its effects upon state variables $sn_j^*$ in target areas.

Hence, basically sensitivities are expressed as $\frac{\delta sn_j^*}{\delta B_{E_i}}$

To derive the analytical expression, we could have used traditional Jacobian-based development for building sensitivities, but observing that sensitivities at system scale depend upon sensitivities at elements level, we rather proceed by building a hierarchy of sensitivities, as illustrated in Fig 3.

In case (A), the sensitivity of the activity level of the pool to perturbation upon its external forcing ($z_E$ or $z_I$) is expressed as function of its sensitivity when uncoupled with itself.

In case (B), the sensitivities of the activity level of each pool to either perturbation upon an external forcing ($x_E$ or $x_I$) are expressed as a function of the uncoupled pools'sensitivities, namely, the expressions found in case (A).

In case (C), the sensitivities of activity level of the excitatory pool of each area upon an external forcing ($B_{E_1}$, $B_{I_1}$, $B_{E_2}$ or $B_{I_2}$) are expressed as a function of the uncoupled area's sensitivities, namely, the expressions found in case (B).

In the three cases, sensitivities at system level are expressed as functions of sensitivities at component level (colored boxes), by opening the feedback loop (cutting colored connections).

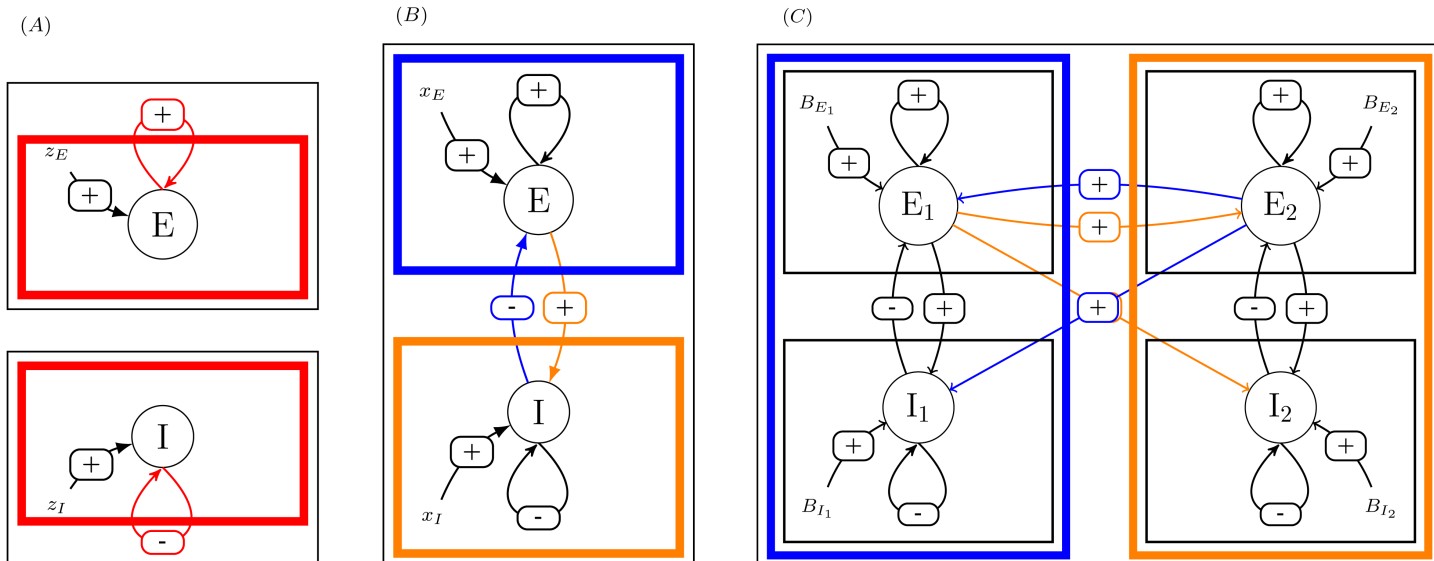

**Fig 3. Principle of the method. (A)** One excitatory pool, either excitatory or inhibitory. **(B)** One area with two coupled pools. **(C)** Two coupled areas.

Starting from the sensitivities of excitatory and inhibitory pools when considered isolated, we can express the sensitivity for one area where the two pools are coupled. At the next level, the sensitivity of coupled areas can be expressed as a function of sensitivities of each area when considered uncoupled. At the next level, the sensitivity for areas belonging to two coupled subsystems can be expressed as functions of sensitivities of areas when considered in their subsystem alone. In the present paper, we will focus on a system consisting of two areas, one control (denoted as $C$) and one target (denoted as $T$), and leave generalization to larger systems to future companion papers (see Perspectives).

Hence, we look for a formal expression for $\frac{\delta sn_T^*}{\delta B_{E_C}}$.

## 2 Models: Building a hierarchy of nested sensitivities

The complete system can be regarded as made of three nested levels: the system itself, which contains areas, which contain pools. Here, we will expose how we build the analytical expressions of the sensitivities at one level as functions of sensitivities at lower level. Since, the principle to build sensitivities at one level from sensitivities at the lower level is generic across levels, we will fully expose this principle in the simplest case (from pool level to area level). For further levels integration, the formal details are given in S2 File and we only report the corresponding results.

### 2.1 At the pool level

Focusing on one isolated pool (here, one excitatory pool, see Fig 3A) at stationary state, its state variables must obey:

$$\begin{cases} 0 = -\beta^E sn^* + \alpha^E T_{glu}(1 - sn^*) rn^* \equiv fn(sn^*, rn^*) \\[2mm] rn^* = \dfrac{a_E xn^* - b_E}{1 - e^{-d_E(a_E xn^* - b_E)}} \equiv hn(xn^*) \\[2mm] xn^* = W_+ J_{nmda} sn^* + z_E \equiv wn(sn^*, z_E) \end{cases} \tag{5}$$

where the *star* notation is to designate fixed point values (below we omit the star notation for the sake of clarity) and $z_E$ represents the external forcing.

To interpret this expression of the system for the isolated excitatory pool, let us first consider the case where $W_+ = 0$. Then, $xn^*$ would be determined solely by the forcing term $z_E$ and would directly determine the corresponding value for $rn^*$. In this case, the dynamics would be linear, and $fn$ would be an affine expression that could be solved analytically.

If now, $W_+ > 0$, then $xn^*$ would also depend upon the value of $sn^*$, so that $rn^*$ would also depend upon $sn^*$, which would in turn affect the value of $sn^*$: hence, from a dynamics perspective, what appears as the support of system non linearity is the self-amplification of the excitatory pool by the recurrent connectivity, which is expressed in the function $wn$ through the parameter $W_+$.

If we now consider a perturbation upon $z_E$, we get a new model, which reads:

$$\begin{cases} fn(sn, rn) = 0 \\ rn = hn(xn) \\ xn = wn(sn, z_E + \delta z_E) \end{cases} \tag{6}$$

and we want the formal expression for the sensitivity $\frac{\delta sn}{\delta z_E}$.

The linearization for the perturbation yields:

$$\begin{cases} \dfrac{\partial fn}{\partial sn}\delta sn + \dfrac{\partial fn}{\partial rn}\delta rn = 0 \\[3mm] \delta rn = \dfrac{dhn}{dxn}\delta xn \equiv hn'\,\delta xn \\[3mm] \delta xn = \dfrac{\partial wn}{\partial sn}\delta sn + \dfrac{\partial wn}{\partial z_E}\delta z_E \end{cases} \tag{7}$$

where the derivatives are to be evaluated at fixed point values (the expression for $hn'$ is given in S2 File).

Plugging the last two equations into the first, we get:

$$\frac{\partial fn}{\partial sn}\delta sn + \frac{\partial fn}{\partial rn}\frac{dhn}{dxn}\left[\frac{\partial wn}{\partial sn}\delta sn + \frac{\partial wn}{\partial z_E}\delta z_E\right] = 0 \tag{8}$$

Rearranging to read how $\delta sn$ depends upon $\delta z_E$:

$$\left(1 + \frac{\partial fn}{\partial sn}^{-1}\frac{\partial fn}{\partial rn}\frac{dhn}{dxn}\frac{\partial wn}{\partial sn}\right)\delta sn = -\frac{\partial fn}{\partial sn}^{-1}\frac{\partial fn}{\partial rn}\frac{dhn}{dxn}\frac{\partial wn}{\partial z_E}\delta z_E \tag{9}$$

This expression is of a primary interest in the methodology to build hierarchical sensitivities: it leads to a *legible* form for identifying how perturbation upon $z_E$ will affect system response $\delta sn$ when in close loop condition with regard to system response $\delta sn^O$ when in open loop condition. Both are related through the feedback gain $g_{sn,z_E}$, that would be identified as:

$$(1 - g_{sn,z_E})\delta sn = \delta sn^O \tag{10}$$

In the absence of recurrent connectivity (setting $W_+ = 0$, hence, *opening the loop* between $sn^*$ and itself), we would have $\frac{\partial wn}{\partial sn} = 0$, so that the only effect of perturbating $z_E$ would translate directly into a perturbation of $sn^*$ in the absence of the feedback loop. This would amount to consider *opening* the feedback loop of $sn$ upon itself and we can denote the open loop response to perturbation as $\delta sn^O$.

Then, the open loop sensitivity is defined as:

$$\mathcal{P}^O_{sn,z_E} \equiv \frac{\delta sn^O}{\delta z_E} = -\frac{\partial fn}{\partial sn}^{-1}\frac{\partial fn}{\partial rn}\frac{dhn}{dxn}\frac{\partial wn}{\partial z_E} \tag{11}$$

Then, we can write the *close loop* sensitivity $\mathcal{P}_{sn,z_E}$ as a function of the *open loop* sensitivity, following:

$$\mathcal{P}_{sn,z_E} \equiv \frac{\delta sn}{\delta z_E} = \frac{1}{1 - g_{sn,z_E}}\mathcal{P}^O_{sn,z_E} \tag{12}$$

where

$$g_{sn,z_E} \equiv -\frac{\partial fn}{\partial sn}^{-1}\frac{\partial fn}{\partial rn}\frac{dhn}{dxn}\frac{\partial wn}{\partial sn} \tag{13}$$

defines the feedback gain for $sn^*$ upon perturbation of $z_E$ through the feedback loop between $sn^*$ and itself. From here, we can analyze the role of closing the loop, depending on the value of $g$, hence predicting the behaviour of the system in response to a perturbation on external input, namely:

- If $g < 0$, then $\frac{1}{1-g} \in ]0; 1[$, so that $\mathcal{P}_{sn,z_E} < \mathcal{P}^O_{sn,z_E}$. Coupling with a negative feedback gain dampens the sensitivity to inputs.

- If $0 < g < 1$, then $\frac{1}{1-g} \in ]0; +\infty[$, so that $\mathcal{P}_{sn,z_E} > \mathcal{P}^O_{sn,z_E}$. Coupling with a positive feedback gain lower than 1 enhances the sensitivity to inputs.

- $g \geq 1$ leads the system in uncharted territories that have to be studied on a per-case basis.

PLOS Computational Biology

In the case of the isolated excitatory pool, the feedback gain is positive by construction (see Fig 4). Finally, we can conclude that (see S2 File for details):

$$\mathcal{P}_{sn,z_E} = \left( \frac{\beta^E + \alpha^E T_{glu} hn(wn(sn^*, z_E))}{\alpha^E T_{glu}(1 - sn^*)hn'(wn(sn^*, z_E))} - W_+ J_{nmda} \right)^{-1} \equiv \varphi n_E(sn^*, z_E) \tag{14}$$

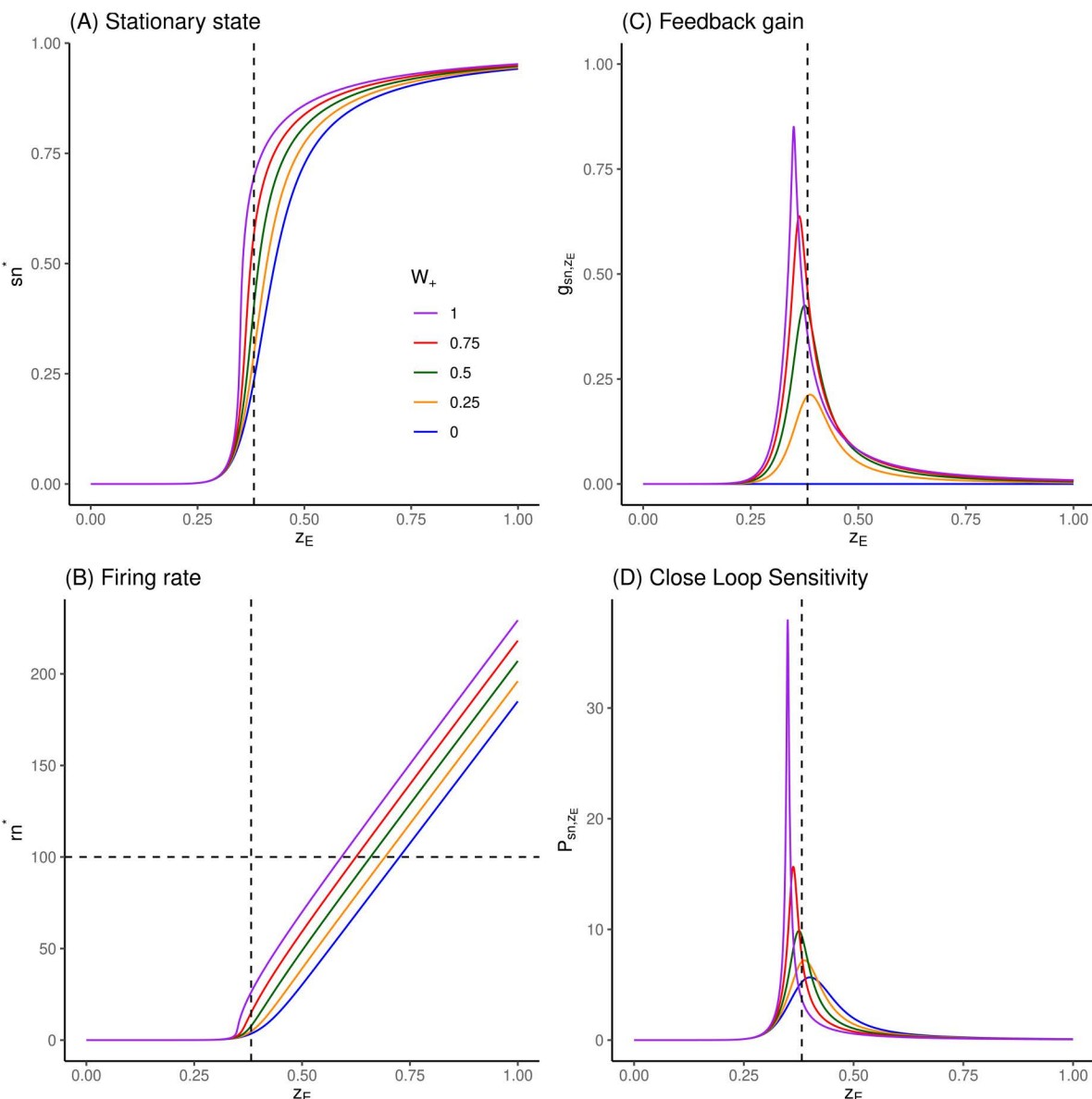

**Fig 4. Response of excitatory pool to inputs, feedback gain and sensitivities. (A)** Fixed point values of $sn^*$ are numerically extracted by root-finding of Eq 5, for values of $z_E \in [0; 1.0]$ and $W_+ \in (0, 0.25, 0.5, 0.75, 1)$. Vertical dashed line correspond to $z_E = x_{E,ref} = 0.382$ nA. **(B)** Corresponding firing rates, following Eq 2. **(C)** Corresponding feedback gain values, following Eq 13. **(D)** Corresponding close loop sensitivities, following Eq 15. See comments in Sec 3.1.

where we define the function $\varphi n_E$ so as to explicitly express the dependencies of the sensitivity for the isolated excitatory pool.

The main point here is to consider that this analytical expression for the sensitivity depends both upon the external forcing, and upon the fixed point under consideration (there can be more than one). The expression will remain the same when we will connect the excitatory pool to the inhibitory one, even if its value will change, since the coupling will modify the values of $sn^*$ and $z_E$ at which to compute it. In the case of the isolated pool, $z_E$ is prescribed as the effective external input, which in turn fixes the fixed point value $sn^*$, whereas in coupled situations, $z_E$ will also depend upon the inputs coming from other parts of the system, which will depend, in turn, upon the stationary values they take in the coupled situation.

Regarding the inhibitory pool, we proceed the same way (see S2 File), and we get:

$$\mathcal{P}_{sg,z_I} = \left( \frac{\beta^I + \alpha^I T_{gaba} hg(wg(sg^*, z_I))}{\alpha^I T_{gaba}(1 - sg^*)hg'(wg(sg^*, z_I))} + J_- \right)^{-1} \equiv \varphi g_I(sg^*, z_I) \tag{15}$$

where function $\varphi g_I$ explicitly express the dependencies of the sensitivity for the isolated inhibitory pool.

In the case of the isolated inhibitory pool, the feedback gain is negative by construction (see Fig 5).

## 2.2 At the area level

Each area contains coupled excitatory and inhibitory pools (see Fig 3B). Here, we are interested in the sensitivity of excitatory and inhibitory pool activity to positive external forcings.

In order to build an analytical expression of these sensitivities, we will consider here again open loop condition versus close loop, now considering that the loop to be opened is the coupling between the two pools.

Open loop sensitivities become the sensitivities of the pools to a perturbation of either $x_E$ or $x_I$ when the connectivity between the two pools is nullified (i.e., in Fig 3B, when the colored connectivity is cut), i.e., the sensitivities to a perturbation upon external forcings (now denoted $x_E$ and $x_I$) when the feedback between the two pools is absent, namely the pool-level sensitivities that have just been built in the previous section.

We denote $\mathcal{A}_{sn,x_E}$ (respectively $\mathcal{A}_{sn,x_I}$) the sensitivity of the excitatory pool activity $sn^*$ to the forcing $x_E$ applied upon the excitatory pool (respectively $x_I$ applied upon the inhibitory pool), and $\mathcal{A}_{sg,x_E}$ (respectively $\mathcal{A}_{sg,x_I}$) the sensitivity of the inhibitory pool activity $sg^*$ to the same forcing. We denote $\mathcal{A}^O_{sn,x_E}$ (respectively $\mathcal{A}^O_{sn,x_I}, \mathcal{A}^O_{sg,x_E}, \mathcal{A}^O_{sg,x_I}$) the corresponding open loop sensitivities.

Following the same lines as for the isolated pools (see S2 File), we express close loop sensitivities (sensitivities for the pools when they are coupled) as functions of open loop sensitivities (sensitivities for the pools with only their recurrent coupling).

Finally, we obtain:

$$\begin{pmatrix} \mathcal{A}_{sn,x_E} & \mathcal{A}_{sn,x_I} \\ \mathcal{A}_{sg,x_E} & \mathcal{A}_{sg,x_I} \end{pmatrix} = \frac{1}{1 + J_{nmda}J_{gaba}\mathcal{A}^O_{sn,x_E}\mathcal{A}^O_{sg,x_I}}$$
$$\times \begin{bmatrix} \mathcal{A}^O_{sn,x_E} & -J_{gaba}\mathcal{A}^O_{sn,x_E}\mathcal{A}^O_{sg,x_I} \\ J_{nmda}\mathcal{A}^O_{sn,x_E}\mathcal{A}^O_{sg,x_I} & \mathcal{A}^O_{sg,x_I} \end{bmatrix} \tag{16}$$

As mentioned above, the main point here is to consider that open loop sensitivities have the same functional form as the isolated pool sensitivities, only they have to be evaluated at the fixed points $sn^*$ and $sg^*$ yielded by the coupled

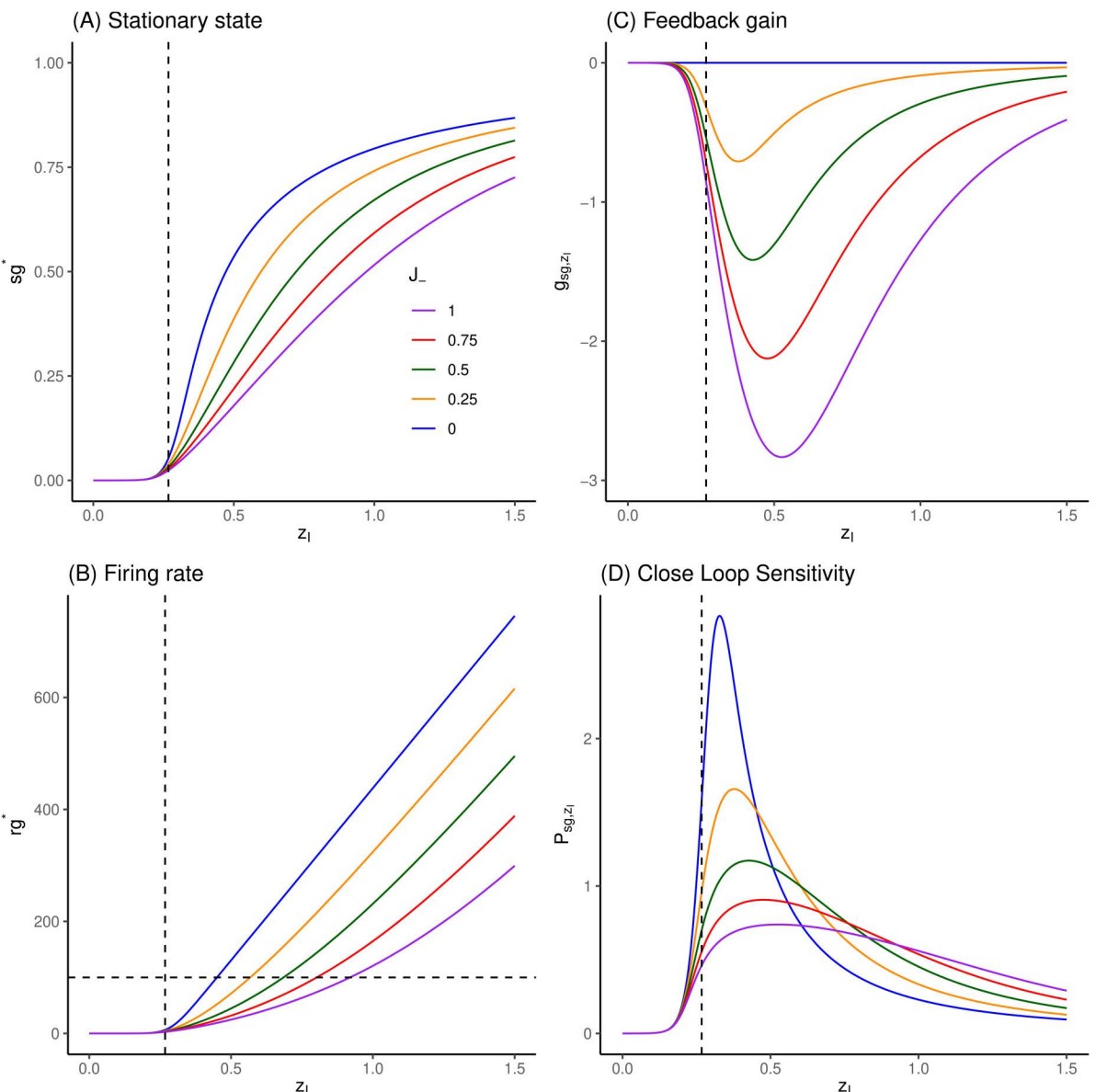

**Fig 5. Response of inhibitory pool to inputs, feedback gain and sensitivities. (A)** Fixed point values of $sg^*$ are numerically extracted by root-finding, for values of $z_I \in [0; 1.5]$ and $J_- \in (0, 0.25, 0.5, 0.75, 1)$. Vertical dashed line correspond to $z_I = x_{I,\text{ref}} = 0.382 * 0.7$ nA. **(B)** Corresponding firing rates, following Eq 2. **(C)** Corresponding feedback gain values. **(D)** Corresponding (close loop) sensitivities, following 15. See comments in Sec 3.1.

system, and considering the total amount of external forcings (i.e., basic forcing $x_E$ or $x_I$ together with the forcing coming from the alternate pool: $z_E = -J_{gaba}sg^* + x_E$ et $z_I = J_{nmda}sn^* + x_I$).

Hence, we can then write the functional forms $\Phi n_E$, $\Phi g_E$, $\Phi n_I$ and $\Phi g_I$ of the sensitivities for an isolated region to explicitly express their dependencies:

$$
\begin{cases}
\mathcal{A}^O_{sn,x_E} &= \varphi n_E(sn^*, -J_{gaba}sg^* + x_E) \\
\mathcal{A}^O_{sg,x_I} &= \varphi g_I(sg^*, J_{nmda}sn^* + x_I) \\
\Phi n_E(\overrightarrow{s^*}, x_E, x_I) &= \dfrac{\mathcal{A}^O_{sn,x_E}}{1 + J_{nmda}J_{gaba}\mathcal{A}^O_{sn,x_E}\mathcal{A}^O_{sg,x_I}} \\
\Phi g_E(\overrightarrow{s^*}, x_E, x_I) &= \dfrac{J_{nmda}\mathcal{A}^O_{sn,x_E}\mathcal{A}^O_{sg,x_I}}{1 + J_{nmda}J_{gaba}\mathcal{A}^O_{sn,x_E}\mathcal{A}^O_{sg,x_I}} \\
\Phi n_I(\overrightarrow{s^*}, x_E, x_I) &= \dfrac{-J_{gaba}\mathcal{A}^O_{sn,x_E}\mathcal{A}^O_{sg,x_I}}{1 + J_{nmda}J_{gaba}\mathcal{A}^O_{sn,x_E}\mathcal{A}^O_{sg,x_I}} \\
\Phi g_I(\overrightarrow{s^*}, x_E, x_I) &= \dfrac{\mathcal{A}^O_{sg,x_I}}{1 + J_{nmda}J_{gaba}\mathcal{A}^O_{sn,x_E}\mathcal{A}^O_{sg,x_I}}
\end{cases}
\tag{17}
$$

where $\varphi n_E$ and $\varphi g_I$ are given by Eq 14 and 15.

## 2.3 At two-area system level

We now turn to the coupling between two areas (see Fig 3C).

In the same way that we have built isolated area sensitivities using functions $\varphi n_E$ and $\varphi g_I$ of isolated pool sensitivities (see (17)), we will use the functions $\Phi n_E$, $\Phi g_E$, $\Phi n_I$ and $\Phi g_I$ defined in (17) to express open loop sensitivities of areas in the two-area system. Hence the loop to be opened is now the connections between the two areas.

In a first step, we have built, in full generality for any two-area system, the expression of the matrix of sensitivities to a perturbation upon the input current in one of the four pools, and they are expressed as functions of the sensitivities in the single-area system (see S2 File).

## 2.4 Control-Target system

In a second step, we use this general expression to address the question of how sensitivities would drive the response of the excitatory pool of one area to the activation of the excitatory pool of the other one, depending on the connectivity between the two areas. Hence, we attribute a role to each area: the area receiving a perturbated input current will be called "Control" area (denoted by C), and the other area will be called "Target area" (denoted by T). Since we are interested in understanding how BOLD signals can be affected by connectivity, we will focus on the expression of the perturbations of their excitatory pools $\delta sn_C$ and $\delta sn_T$ (which drive changes in BOLD signals, see [33]) in response to a perturbation $\delta B_{E_C}$ applied to the input current of Control Area.

We obtain (see S2 File):

$$
\begin{pmatrix} \delta sn_C \\ \delta sn_T \end{pmatrix} = \frac{1}{1 - G_{CT}G_{TC}} \begin{pmatrix} \mathcal{A}_{sn_C, x_{E_C}} \\ G_{TC}\mathcal{A}_{sn_C, x_{E_C}} \end{pmatrix} \delta B_{E_C}
\tag{18}
$$

with

$$
\begin{cases}
G_{CT} = \mathcal{A}_{sn_C, x_{E_C}} k_{E_{CT}}\kappa_{CT} + \mathcal{A}_{sn_C, x_{I_C}}(1 - k_{E_{CT}})\kappa_{CT} \\
G_{TC} = \mathcal{A}_{sn_T, x_{E_T}} k_{E_{TC}}\kappa_{TC} + \mathcal{A}_{sn_T, x_{I_T}}(1 - k_{E_{TC}})\kappa_{TC}
\end{cases}
\tag{19}
$$

Written in the *legible* form, we then have for Control area:

$$
(1 - G_{CT}G_{TC})\delta sn_C = \mathcal{A}_{sn_C, x_{E_C}}\delta B_{E_C}
\tag{20}
$$

so we can define $g_{sn_C, B_{E_C}} = G_{CT}G_{TC}$ the feedback gain for $\delta B_{E_C}$ acting on $\delta sn_C$ through the feedback loop, from Control to Control area via Target area.

For Target area, we have:

$$(1 - G_{CT}G_{TC})\delta sn_T = G_{TC}\mathcal{A}_{sn_C, x_{E_C}}\delta B_{E_C} \tag{21}$$

so that the feedback gain $g_{sn_T, B_{E_C}}$ acting on $\delta sn_T$ is the same, namely:

$$g_{sn_{C|T}, B_{E_C}} = \left(\mathcal{A}_{sn_C, x_{E_C}} k_{E_{CT}}\kappa_{CT} + \mathcal{A}_{sn_C, x_{I_C}}(1 - k_{E_{CT}})\kappa_{CT}\right)$$
$$\times \left(\mathcal{A}_{sn_T, x_{E_T}} k_{E_{TC}}\kappa_{TC} + \mathcal{A}_{sn_T, x_{I_T}}(1 - k_{E_{TC}})\kappa_{TC}\right) \tag{22}$$

## 2.5  Effect of connectivity upon Target sensitivity

So far, we have derived the formal expressions for sensitivities and gains for any connectivity, i.e., for any values of $k_{E_{ij}}$ that governs how excitatory and inhibitory pools are connected between areas. We will now consider archetypal possibilities by assigning binary values to $k_{E_{CT}}$ and $k_{E_{TC}}$.

Considering that (long-range) connexions always originate from excitatory pools, we then have four possibilities, depending for each area on whether it receives excitatory signal upon its excitatory pool or its inhibitory one (see Fig 6).

We can then produce predictions on system behavior for the schemes of connectivity representing inhibitory control models: Target-to-Control inhibition (I-E), mutual inhibition (I-I) or Control-to-Target inhibition (E-I), w.r.t. the "model of reference" which is mutual excitation (E-E).

We convene to denote the connectivity by □ – □ where □ denotes the reception pools (*E* or *I*) for the Control and Target areas respectively.

Accordingly, we will denote the corresponding feedback gain $g_{sn_T, B_{E_C}}$ as: $g_{\square\text{-}\square}$.

We can then write a generic expression of feedback gains and sensitivities for any connectivity:

$$\begin{cases} g_{\square\text{-}\square} = \mathcal{A}_{sn_C, x_{\square_C}}\kappa_{CT}\mathcal{A}_{sn_T, x_{\square_T}}\kappa_{TC} \\[2ex] \delta sn_C = \dfrac{\mathcal{A}_{sn_C, x_{E_C}}}{1 - g_{\square\text{-}\square}}\delta B_{E_C} \\[2ex] \delta sn_T = \dfrac{\mathcal{A}_{sn_T, x_{\square_T}}\kappa_{TC}\mathcal{A}_{sn_C, x_{E_C}}}{1 - g_{\square\text{-}\square}}\delta B_{E_C} = \mathcal{A}_{sn_T, x_{\square_T}}\kappa_{TC}\delta sn_C \end{cases} \tag{23}$$

yielding the sensitivities given in Table 1 for each connectivity.

From this, we can now predict the signs of sensitivities:

- For $g_{\square\text{-}\square} < 1$, since $\mathcal{A}_{sn_C, x_{E_C}} > 0$ (see section 3.2), we always have $\delta sn_C > 0$.

- Sensitivities for Target area are given by:

| □ – □ | $\mathcal{A}_{sn_T, x_{\square_T}}$ | $\delta sn_T$ |
|-------|------------------------------------|---------------|
| $E-E$ | $\mathcal{A}_{sn_T, x_{E_T}} > 0$ | >0 |
| $I-E$ | $\mathcal{A}_{sn_T, x_{E_T}} > 0$ | >0 |
| $I-I$ | $\mathcal{A}_{sn_T, x_{I_T}} < 0$ | <0 |
| $E-I$ | $\mathcal{A}_{sn_T, x_{I_T}} < 0$ | <0 |

We conclude that only I-I or E-I connectivity could translate upregulation in Control area into downregulation in Target area.

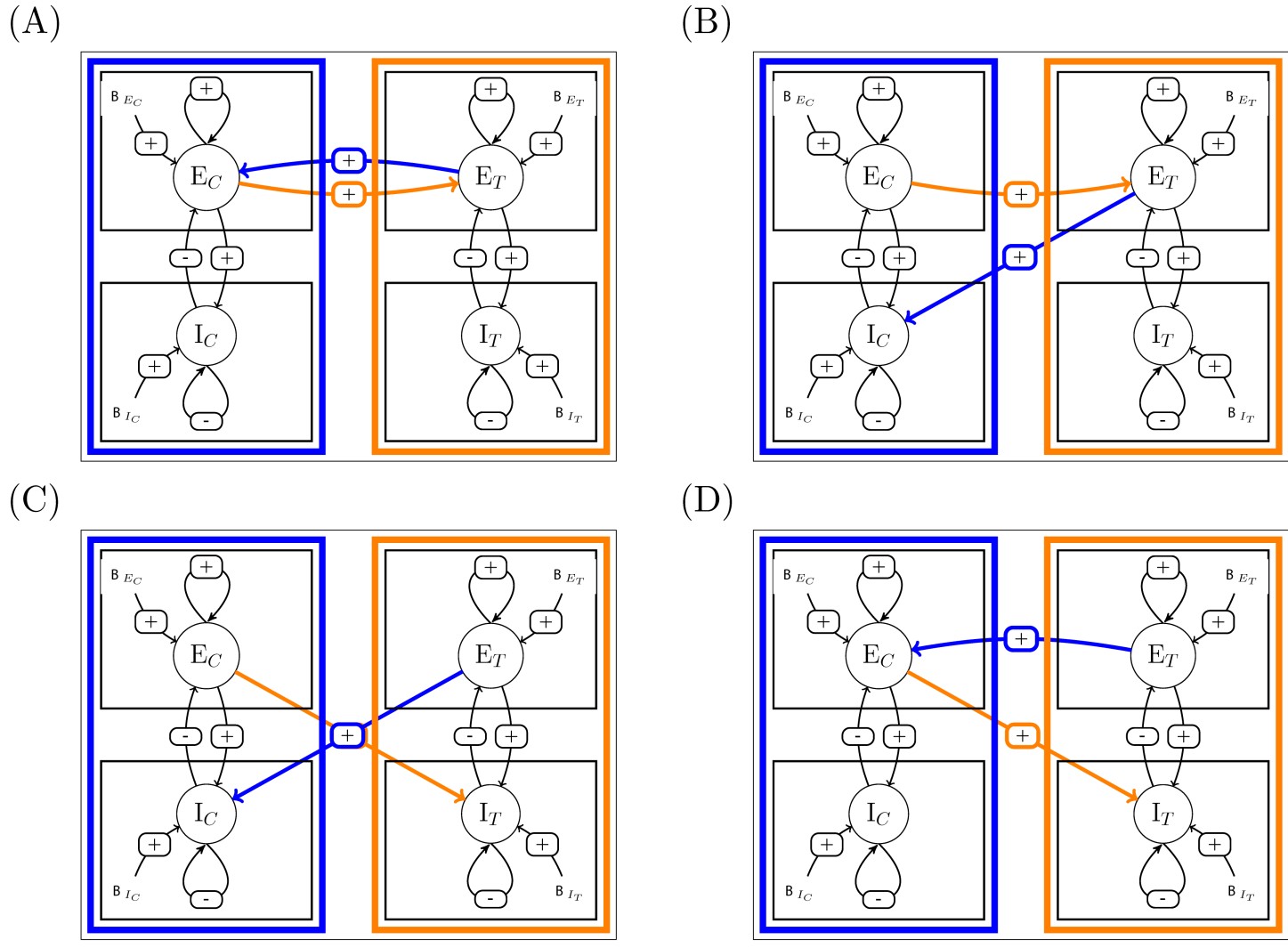

**Fig 6. Connectivity that are considered. (A)** Mutual excitation **(E-E)**, **(B)** Target-to-Control inhibition **(I-E)**, **(C)** Mutual inhibition (I-I), **(D)** Control-to-Target inhibition (E-I). We denote the connectivity by □ − □ where first □ denotes the reception pools (*E* or *I*) for the Control (in blue) and second one for the Target (in orange) areas respectively.

**Table 1. Sensitivities for $sn^*$ in Control and Target area to perturbation upon $B_{E_C}$, the input current into Control Area.**

| □ − □ | $g_{□-□}$ | $\delta sn_C$ | $\delta sn_T$ |
|---|---|---|---|
| $E-E$ | $\mathcal{A}_{sn_C,x_{E_C}}\kappa_{CT}\mathcal{A}_{sn_T,x_{E_T}}\kappa_{TC}$ | $\frac{\mathcal{A}_{sn_C,x_{E_C}}}{1-g_{E-E}}\delta B_{E_C}$ | $\mathcal{A}_{sn_T,x_{E_T}}\kappa_{TC}\delta sn_C$ |
| $I-E$ | $\mathcal{A}_{sn_C,x_{I_C}}\kappa_{CT}\mathcal{A}_{sn_T,x_{E_T}}\kappa_{TC}$ | $\frac{\mathcal{A}_{sn_C,x_{E_C}}}{1-g_{I-E}}\delta B_{E_C}$ | $\mathcal{A}_{sn_T,x_{E_T}}\kappa_{TC}\delta sn_C$ |
| $I-I$ | $\mathcal{A}_{sn_C,x_{I_C}}\kappa_{CT}\mathcal{A}_{sn_T,x_{I_T}}\kappa_{TC}$ | $\frac{\mathcal{A}_{sn_C,x_{E_C}}}{1-g_{I-I}}\delta B_{E_C}$ | $\mathcal{A}_{sn_T,x_{I_T}}\kappa_{TC}\delta sn_C$ |
| $E-I$ | $\mathcal{A}_{sn_C,x_{E_C}}\kappa_{CT}\mathcal{A}_{sn_T,x_{I_T}}\kappa_{TC}$ | $\frac{\mathcal{A}_{sn_C,x_{E_C}}}{1-g_{E-I}}\delta B_{E_C}$ | $\mathcal{A}_{sn_T,x_{I_T}}\kappa_{TC}\delta sn_C$ |

# 3 Results and discussion

Numerical illustrations of these formal developments are given below using the parameters values of reference listed in S1 File. All codes needed to build these numerical illustrations using Python 3 language are given in S3 File.

## 3.1 Effect of self-coupling in pools

To illustrate the behavior of the basic unit of the system, the effect of incoming input $z_E$ upon the activity level $sn^*$ in an isolated excitatory pool is illustrated in Fig 4 for increasing values of its self-excitatory coupling parameter $W_+$.

As $z_E$ increases, fixed points $sn^*$ increase monotonically up to the saturating value 1 (Fig 4A). With the set of parameters given in [1] (listed in 5), we observe a threshold effect, with an absence of reaction to input values lower than about $z_E \simeq 0.3$ nA. This threshold effect is due to the filtering effect in the translation from input to $sn$ through firing rate $rn$ (Eq 2, Fig 4B), so that if $rn^* \to 0$ in Eq 5, so must do $sn^*$.

Beyond this threshold, fixed points values increase strongly with small increments of $z_E$, such that the range of relevant input values is pretty narrow. If we consider a firing rate at 100 Hz as the reasonable maximal value for a highly activated pool, this range would span $[0.3; 0.75]$ nA in absence of self-excitatory loop ($W_+ = 0$) down to $[0.3; 0.6]$ nA for $W_+ = 1$ (Fig 4B).

This high reactivity to input is explained by the feedback gain (Fig 4C), which is positive by construction and can get strong values as the self-excitatory coupling parameter $W_+$ is increased. When $W_+ = 0$, the gain is analytically null (blue curves), and the corresponding sensitivity is the open loop one (the lowest one in Fig 4D). As $W_+$ increases, the gain remains null for $sn^* \to 0$ corresponding to $z_E < 0.3$ (since there is no input for self-amplification), and also saturates towards 0 for $sn^*$ saturating to 1 (since there is no more possible gain). In between, the gain reaches a maximal value in the range of $z_E$ which has a maximal impact upon $sn^*$. The close loop sensitivity is then scaled accordingly.

The effect of incoming input $z_I$ upon the activity level $sg^*$ in an isolated inhibitory pool is illustrated in Fig 5 for increasing values of its self-inhibitory coupling parameter $J_-$. As expected, increasing $z_I$ increases fixed points $sg^*$ values monotonically, but here, at a far lower rate than in excitatory pool (Fig 5A). Since the self-coupling is inhibitory, the feedback gain is negative (Fig 5C), hence the sensitivity is the highest in absence of it (blue curves, Fig 5D). As a consequence, the self-inhibitory coupling will extend the range of relevant input values $z_I$ for firing rates below 100 Hz (Fig 5B), from about $[0.2; 0.45]$ nA to about $[0.2; 0.9]$ nA.

## 3.2 Effect of coupling two pools

On Figs 4A and 5A, vertical dashed lines report the values representing *basic external inputs* $x_E$ and $x_I$ when the pools are connected to form an area, and that are given in [1]. From their location in the range of relevant values, we can anticipate the role of the inhibitory pool upon the basic level of activity in one isolated area: since $x_I$ is close to the threshold, it will be up-regulated by the excitatory input from the excitatory pool, so that its induced inhibitory input backward to the excitatory pool will actually lower $z_E$ towards values near the threshold. Namely, with the value of $x_E$ prescribed by [1], the excitatory pool *needs* to be down-regulated to have reasonable excitation level at the resting state.

This effect of coupling the two pools is illustrated in Fig 7, where we submit the area to varied values of $x_E$ or $x_I$ around the values of references (vertical dashed lines) and for a range of $J_{gaba}$ which controls the strength of feedback inhibition from inhibitory pool to excitatory pool upon activation of the latter. This feedback inhibition then acts as a self-inhibition at the area level.

We report on the left column how varying $x_E$, while keeping $x_I$ at the reference value, affects the activity level of the excitatory pool. In this case, increasing $x_E$ will, on the one hand, directly enhance activity level in the excitatory pool, on the other hand, this enhanced activity level will in turn enhance activity level in the inhibitory pool (by feedforward activation), which will in turn exert a feedback inhibition effect upon the excitatory pool. All feedbacks taken into account

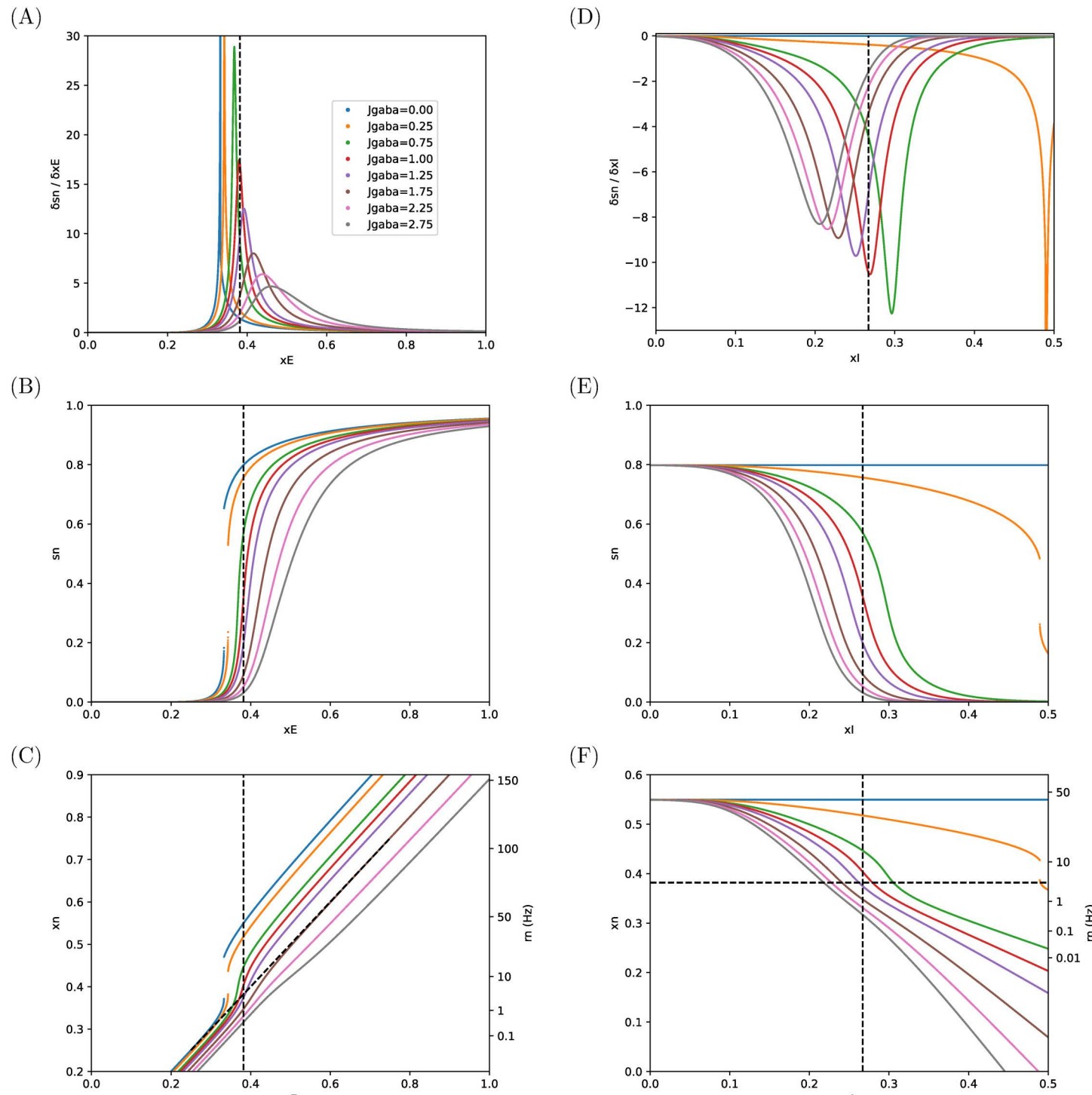

**Fig 7. Sensitivities and responses of excitatory pool in an isolated area. (A—C)** Response to external input upon the excitatory pool, **(D—F)** Response to external input upon the inhibitory pool. **(A, D)** Sensitivities, **(B, E)** $sn^*$, **(C, F)** $xn^*$. See comments in Sec 3.2.

(self-excitation and feedback inhibition), the sensitivity $\mathcal{A}_{sn,x_E}$ remains positive (Fig 7A), albeit with a responsiveness that can be far smoother than for the isolated pool.

We first note that sensitivities have here again bell-shape curves. Indeed, for lower values of external input ($x_E < 0.25$), $sn^*$ values would slowly tend to 0, whereas for too high values of external input ($x_E > 0.6$) it would slowly saturate to 1 (Fig 7B). In between, sensitivities must have a peak value.

The responsiveness to external inputs depends upon the strength of the self-inhibition at the area level, which is governed by $J_{gaba}$. For lower values of $J_{gaba}$, the responsiveness is steep, and the range of operational values of total input remains narrow. As a matter of fact, in the case $J_{gaba} = 0$, we recover the sensitivity of the isolated excitatory pool (Eq 14), yet for the high value $W_+ = 1.4$ that is prescribed when coupling pools in [1], and we observe a divergence of the sensitivity around $x_E \simeq 0.33$, which translates into a jump for $sn^*$ in Fig 7B.

This means that, in absence of a sufficient down-regulation of the excitatory pool by the feedback inhibition, the slightest variation in total input could translate into an all-or-nothing response, which is not a property we would consider as reasonable for a neural pool. This divergence appears no more for $J_{gaba}$ greater than 0.75 nA.

As $J_{gaba}$ value becomes larger and larger, the responsiveness is smoother and smoother as the peak of sensitivity to inputs tends to decrease, and the operational range of input values increases. Since sensitivities can be regarded as the derivatives of $sn^*$ with respect to the input, their bell-shaped curves translate into *sigmoidal* shapes when considering $sn^*$ (Fig 7B) and the enlargement of the operational range translates into a response of the pool that becomes gradual.

Depending on $J_{gaba}$, the $sn^*$ level at the value of reference for $x_E$ spreads over quite large a range (from about 0.6 for $J_{gaba} = 0.75$ down to about 0.05 for $J_{gaba} = 2.25$).

This spread in the excitatory pool $sn^*$ is however largely reduced when considering the total effective input $xn^*$, as it is tempered by the negative contribution from the inhibitory input. For instance, in the case $J_{gaba} = 0.75$ where $sn^* \simeq 0.6$, the total input current is about $xn^* \simeq 0.45$ (Fig 7C) and would translate into a limited firing rate ($rn^* \simeq 16$ Hz whereas it would be $rn^* = 45$ Hz for $J_{gaba} = 0$).

Actually, for large enough a value for $J_{gaba}$ (around 1.75nA), inhibitory feedback control at the area level would compensate exactly for self-excitatory feedback in the excitatory pool (Fig 7C, where oblique dotted line corresponds to $xn^* = x_E$). With such a high value of $J_{gaba}$, an external input $x_E = 0.75$ nA would be fully counter-balanced and yield a firing rate at 100 Hz, as if in an isolated excitatory pool with no self-excitation ($W_+ = 0$, blue curve in Fig 4B).

On the right column of Fig 7, we report the symmetrical effect of varying $x_I$ while keeping $x_E$ at the value of reference. The sensitivity $\mathcal{A}_{sn,x_I}$ of the activity ($sn^*$) in the excitatory pool with regards to positive perturbation upon input current into the inhibitory pool is, as expected, negative (Fig 4D) since increasing input current into the inhibitory pool will lower the input current into the excitatory pool (Fig 4F). We note that in absence of external input current into the inhibitory pool ($x_I = 0$), we have an almost null activity in the inhibitory pool (i.e., $sg^* \simeq 0$), meaning that the forward excitation from the excitatory pool is not enough *per se* to activate it (with the current set of parameters) so that we recover the high stationary state of the isolated excitatory pool ($sn^* = 0.8$, Fig 7E).

On the other hand, with too strong an activation of the inhibitory pool ($x_I > 0.4$), $sn^*$ would be flattened to 0 and loose any responsiveness.

In between, the shape of sensitivities curves are poorly affected by $J_{gaba}$ (for $J_{gaba} \geq 0.75$) so that the range of operational values for $x_I$ remains about the same, yet with a median value which is lower and lower as $J_{gaba}$ is increased.

In Fig 7F, vertical dashed line represents $x_{I,ref} = 0.382 \times 0.7$ nA and horizontal dashed line represents the cases $x_n = x_{E,ref} = 0.382$ nA, corresponding to situations when inhibitory feedback compensates self-excitation. When $xn < x_{E,ref}$ (below horizontal dashed line), inhibitory feedback over compensates self-excitation. In the opposite case, inhibitory feedback only acts as a brake on self-excitation. A crossing value between the two behaviours appears to be around $J_{gaba} = 1.25$ nA. An important point is then that, with $J_{gaba} \geq 1.25$, any additional input current to the inhibitory pool will translate into a decrease of the total input current into the excitatory pool (i.e., $xn < x_{E,ref}$).

To summarize, we will point out at four main points:

1. The sensitivities to positive perturbation of input current to either excitatory ($\mathcal{A}_{sn,x_E}$) or inhibitory pool ($\mathcal{A}_{sn,x_I}$) are respectively positive and negative. This point will be of importance for the interpretation of the Control-Target system behaviour.

2. At state of reference (i.e., setting $x_E$ and $x_I$ at their values of reference representing *basic inputs* with no additional input from other areas), the stationary values of the excitatory pool ($sn^*$ and $rn^*$) can be directly adjusted by tuning $J_{gaba}$.

3. With the set of parameters prescribed by [1], the responsiveness of the area can be steep when the inhibitory feedback, driven by $J_{gaba}$ from the inhibitory pool to the excitatory pool upon positive perturbation of the latter, is too low. Moreover, even with a high strength of the self-inhibitory feedback loop within the area, the operational range for further inputs is not so large: with the given values of references, higher values of $J_{gaba}$ would at most allow a doubling of input current with regards to basic input current.

4. Given the structure of the model, activation of inhibitory pool by external input can either translate into a lower excitation of the excitatory pool (a brake on self-excitation), or into a inhibition of the excitatory pool (inhibition becomes greater than self-excitation), depending upon how the excitation of the inhibitory pool can have an effect upon total current input into the excitatory pool through $J_{gaba}$. A crossing value between the two behaviors appears to be around $J_{gaba}$ = 1.25 nA.3.3 Effect of connectivity upon sensitivities in Control-Target system

We now turn to illustrations of the Control-Target system's behaviour, depending upon the connectivity, as described in Fig 6.

In all cases, $J_{gaba_C}$ in the Control area and $J_{gaba_T}$ in the Target area are set such that the firing rate $rn_C = rn_T$ = 3 Hz for the input currents of reference, i.e., at rest. For an area connected by pool E $J_{gaba_{C|T}}$ = 1.758 nA and for an area connected by pool I $J_{gaba_{C|T}}$ = 0.929 nA. Then we study the effect of varying the excitatory input current $B_{E_C}$ into the Control area, for a range of $J_{gaba_T}$ in the Target area. Note that changing $J_{gaba_T}$ while keeping $J_{gaba_C}$ affects the firing rates at rest.

In all cases, we verify that the signs of sensitivities (see Figs 8C, 8D, 9C, 9D, 10C, 10D, 11C and 11D) are the same as those predicted in section 2.5.

In E-E connectivity (mutual excitation, Fig 8), the excitatory pools are in a mutual excitation regime (Fig 8A) so that both sensitivities to perturbation upon $B_{E_C}$ are positive (Figs 8C and 8D), and Target activity can only increase upon Control excitation. Moreover, the feedback gain associated with the inter-area connection is positive (Fig 8B). Upon stimulation of Control, this positive feedback will then amplify the effect of the stimulation, and stabilize the activity levels of both areas at higher levels than in absence of mutual excitation. The capacity of Control area to increase Target activity is modulated by the value set for $J_{gaba_T}$ (Figs 8E and 8F). For a low value of $J_{gaba_T}$ (blue curve in Fig 8F), the self-inhibition at Target area level is less operative so that excitation of Control can translate in higher activity in Target (here illustrated by firing rates). We note however that the capacity to increase Target activity can be limited to narrow ranges of $B_{E_C}$. For example, for $J_{gaba_T}$ = 1.75 nA, the effective range of control is about [0.35, 0.425], i.e., a 20% increase, to take firing rate in Target from about 2 Hz to about 15 Hz. Beyond this interval, further increase of Control will have poor effect (e.g., doubling the firing rate in Control, from 25 Hz to 50 Hz, will translate into an additional 5 Hz in Target). Strikingly, sensitivities peak values, as core values of range for effective control, are quite disparate, so that control efficacy would highly depend upon the value set for basic input current into Control area, which should become a function of $J_{gaba_T}$.

In I-E connectivity (Target-to-Control inhibition, Fig 9), the Control area has an excitatory effect upon Target area, which in turn has an inhibitory effect upon Control area. Here again, both sensitivities to perturbation upon $B_{E_C}$ are positive (Figs 11C and 11D), and Target activity can only increase upon Control excitation. However, the feedback gain associated with the inter-area connection becomes negative (Fig 11B) so that the effect of the Control stimulation will be dampened in

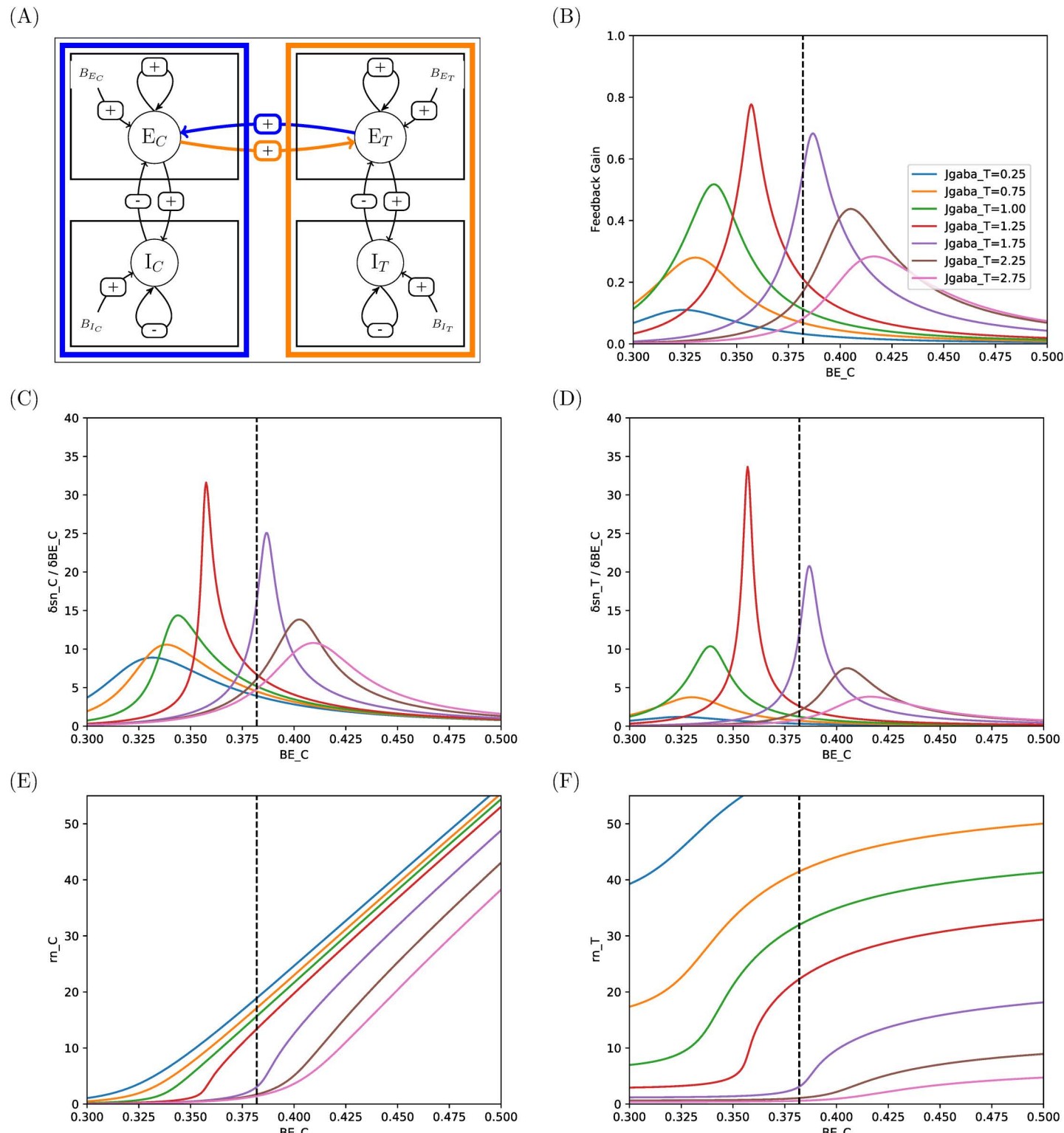

**Fig 8. Responses and sensitivities of excitatory pools in two-area system with connectivity E-E. (A)** Illustration of the connectivity. **(B)** Feedback gain $g_{sn_{C|T},B_{E_C}}$ (Eq 22). **(C, D)** Sensitivities of $sn$ to perturbations upon excitatory pool of Control Area, in Control and Target areas respectively. **(E,F)** Corresponding firing rates. See comments in Sec 3.3.

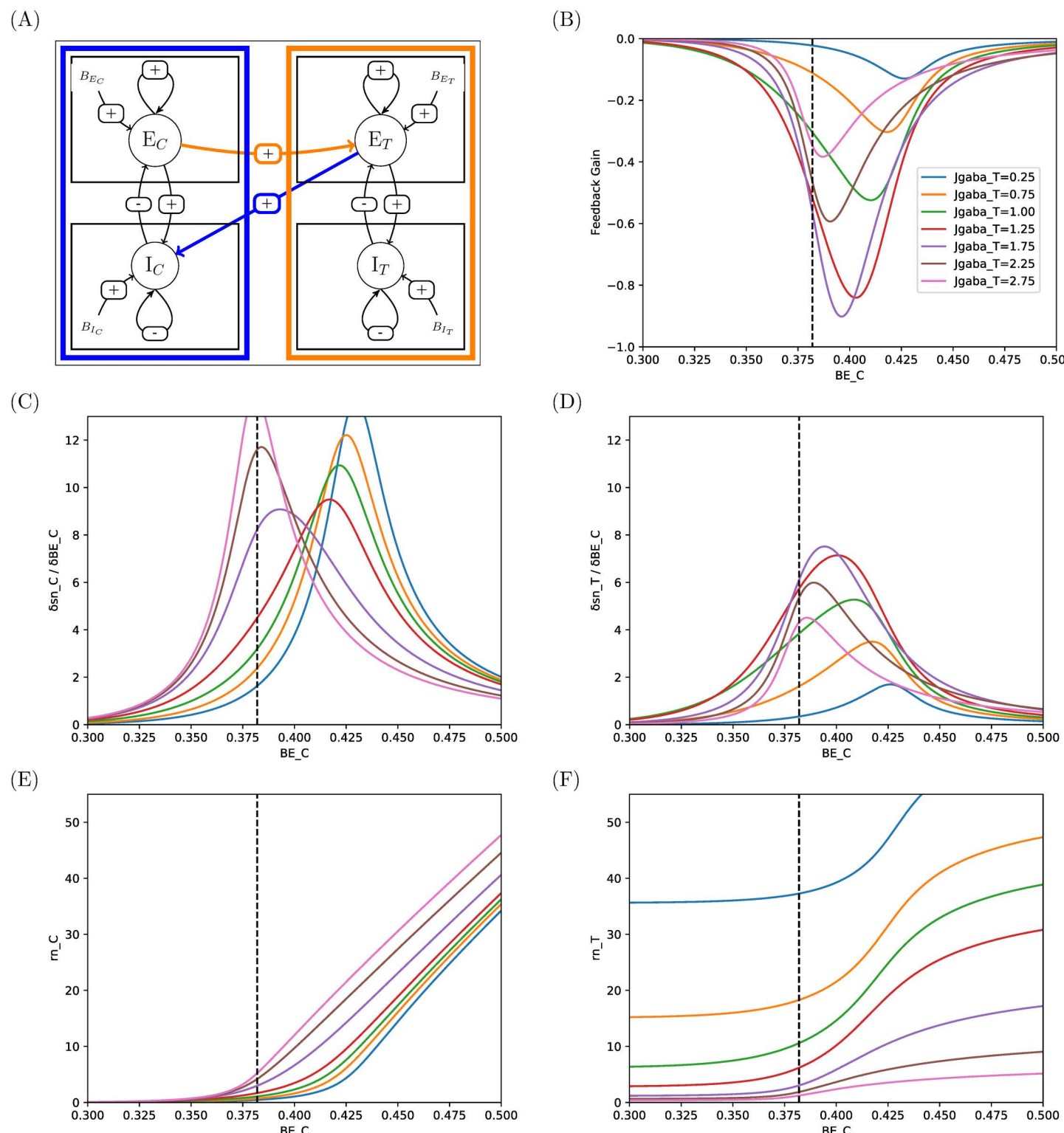

**Fig 9. Responses and sensitivities of excitatory pools in two-area system with connectivity I-E.** Same legend as in Fig 8. See comments in Sec 3.3.

both areas, resulting in activity levels lower than in absence of the feedback loop. The capacity of Control area to increase Target activity is still modulated by the value set for $J_{gaba_T}$ (Figs 11E and 11F), but, by contrast with E-E connectivity, the range of effective control is enlarged and become pretty similar across $J_{gaba_T}$ values (Figs 11C and 11D), so that changing $J_{gaba_T}$ mainly regulates the range for firing rates in presence versus in absence of Control activation.

Direct projection of Control area to inhibitory pool of Target area (I-I and E-I connectivity) completely changes the picture: activating Control can now *decrease* activity in Target area, because sensitivities of $sn_T$ to perturbation upon $B_{E_C}$ become negative in both cases.

In I-I connectivity (mutual inhibition, Fig 10), the feedback gain is positive and sensitivities can reach high values and span narrow ranges of $B_{E_C}$, as in E-E connectivity (note that the case $J_{gaba_T} = 0.75$ nA yields a jump for $B_{E_C}$ close to 0.4 nA and can be disregarded), here again yielding narrow ranges for control. For instance, for the condition ensuring resting state at 3 Hz in both areas for basic input current set to their value of reference ($J_{gaba_T} = J_{gaba_C} = 0.929 \simeq 1$ nA, hence near the green curve in Fig 10F), a slight activation of the Control area by $\triangle B_{E_C} = 0.425 - x_{E,\text{ref}} = 0.043$ nA, i.e., a small 10% increase, would be sufficient to shut the activity ot Target down to 0.5 Hz.

In E-I connectivity (Control-to-Target inhibition, Fig 11), by contrast, sensitivities of Target to activation of Control area span the full range considered for $B_{E_C}$ and their peak values are quite lower (in absolute value), so that effect of Control is smoother and can be nicely gradual (as in I-E connectivity). For instance, for the 3 Hz condition ($J_{gaba_C} = 1.758$ and $J_{gaba_T} = 0.929$, green curves), there would need now about $\triangle B_{E_C} = 0.5 - x_{E,\text{ref}} = 0.118$ nA in Control to get 0.5 Hz in the Target, i.e., a 30% increase.

Note that in both cases however, the 10% increase in I-I and the 30% increase in E-I, the activity level in Control area would shift from 3 Hz to about the same 30 Hz.

To summarize, there are here two main points:

1. In mutual excitation E-E, the upregulation of Control is amplified by the feedback loop in both areas; in Target-to-Control inhibition I-E, the upregulation of Control is dampened in both areas; in mutual inhibition I-I, the feedback loop amplifies the downregulation of Target triggered by upregulation of Control, and in Target-to-Control inhibition E-I, it is dampened,

2. In I-E and E-I connectivity, control effectivity is more gradual with regards to $J_{gaba_T}$ than in E-E and I-I.

### 3.4 Consequences for system responses upon step activation of Control area

So far, we have analyzed the response of the system in terms of sensitivities, namely how $sn_C$ and $sn_T$ would be affected by an infinitesimal (positive) perturbation applied upon the excitatory forcing of the excitatory pool of Control area. Formal expression of sensitivities are the only path to derive analytical results that allow to understand the dynamical behavior of the system in full generality.

We now turn to illustrations that extend this analysis by direct comparison between states of the system at rest versus when submitted to a finite step of activation, as it is a typical case in macroscopic imagery such as fMRI, where the BOLD signal is recorded during tasks like Go-NoGo or Think-NoThink in inhibitory control studies ([19,32]).

**3.4.1 Illustrations with BOLD signals.** In Fig 12, we illustrate the system dynamical response for E-E (left) and E-I (right) connectivities, when submitted to a step stimulation of $\triangle B_{E_C} = 0.1$ nA upon the excitatory pool of Control area. We report the behavior of synaptic activities $sn(t)$ and of the BOLD signal in both areas, when submitted to a fictitious experiment with a 20s long stimulation (upper row) or a more realistic experiment (as, e.g., in [32]) with four 3s long stimulations interspaced by random delay (lower row).

As expected from Fig 2, the dynamical response of the synaptic activity to stimulus onset is step-like in all case, and, as expected from our sensitivities analysis, the Target responds by a decrease of activity (both in synaptic activity, and then in BOLD signal) in E-I connectivity.

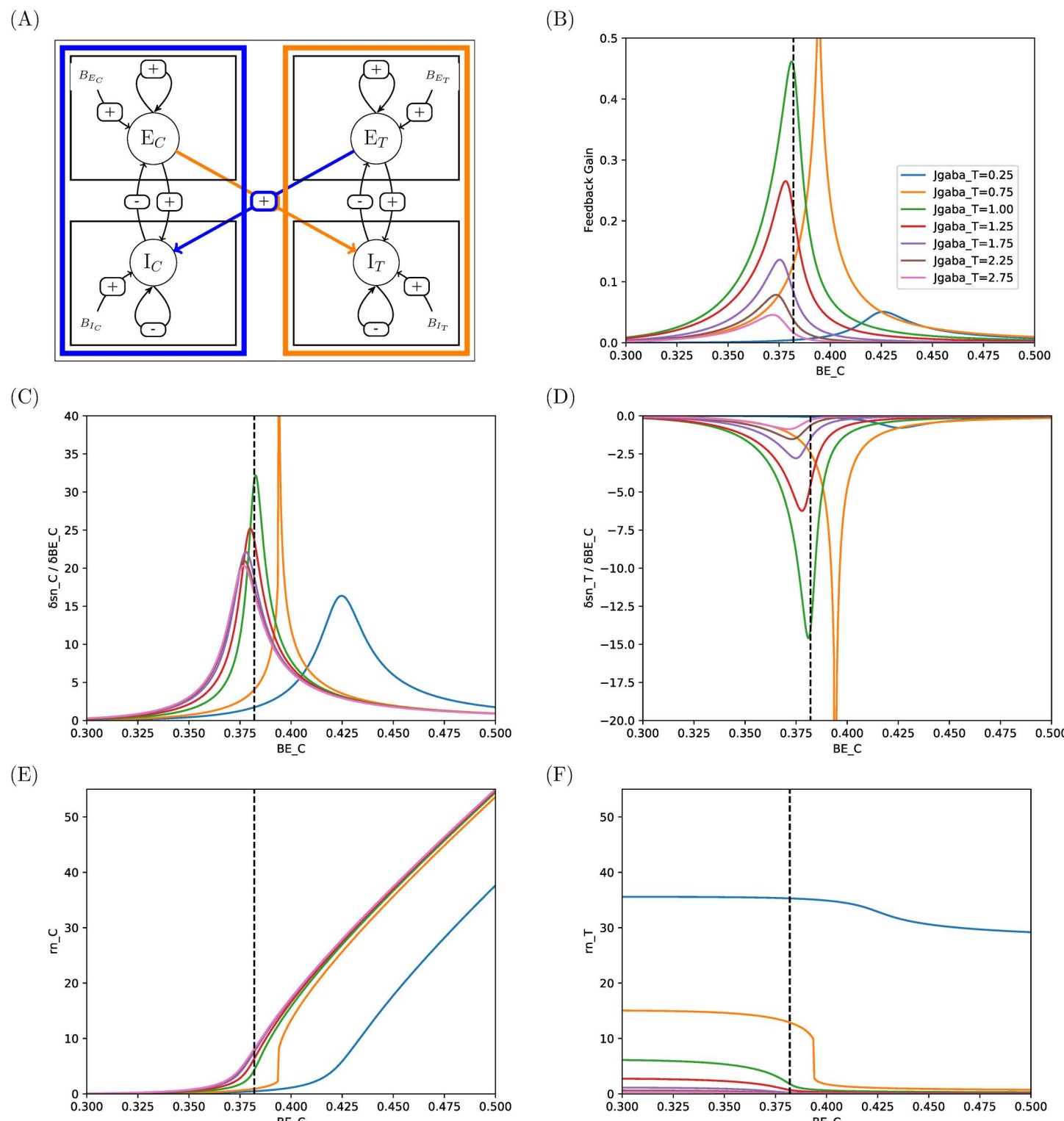

**Fig 10. Responses and sensitivities of excitatory pools in two-area system with connectivity I-I.** Same legend as in Fig 8. See comments in Sec 3.3.

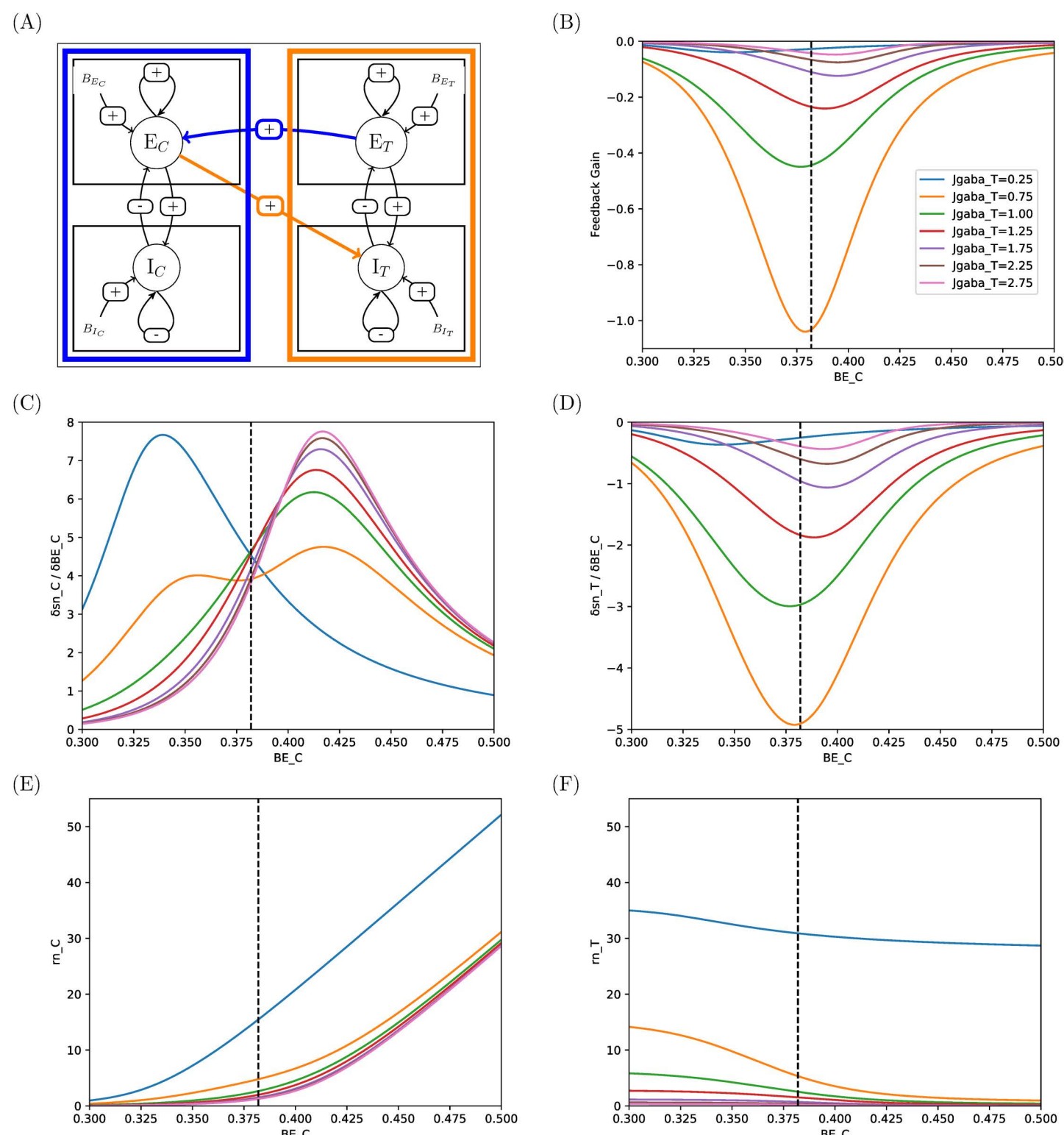

**Fig 11. Responses and sensitivities of excitatory pools in two-area system with connectivity E-I.** Same legend as in Fig 8. See comments in Sec 3.3.

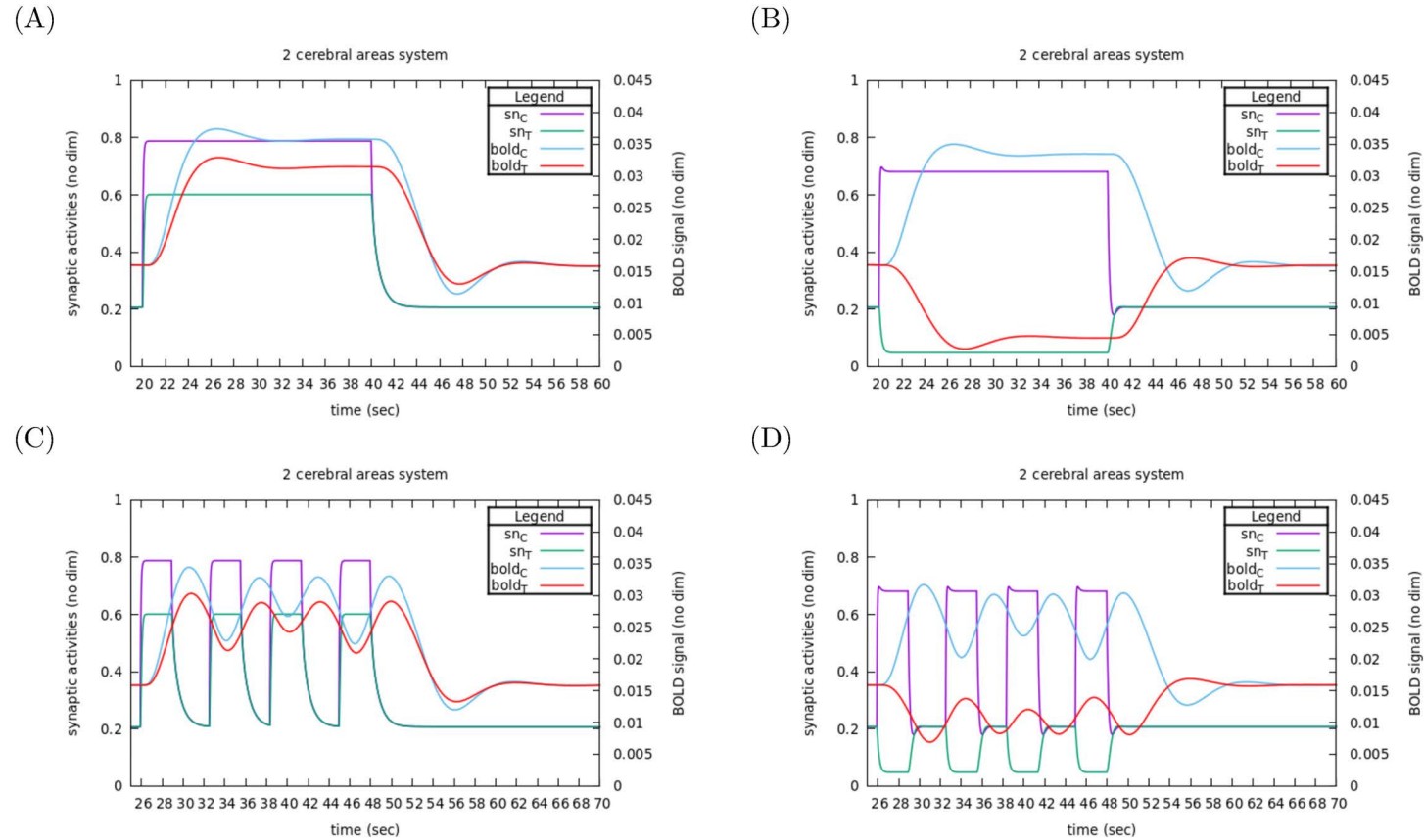

**Fig 12. Response as BOLD signals in E-E and E-I connectivities.** Left: E-E connectivity, right E-I connectivity. (A, B) A stimulation of 0.1 nA is applied upon the excitatory pool of the Control area during 20 s. (C, D) Time course of stimulations are made typical of a TNT task with stimulation steps during 3 s, separated by random intervals between 2.4 and 3.6 s. See comments in Sec 3.4.1.

The long stimulation gives an idea of the characteristic times of the dynamics. Contrasting to synaptic activities, the BOLD signal needs about 15s to reach its stimulus-induced steady state value and about the same delay to return to its rest value after the stimulus stop.

If, in both connectivities, synaptic activities adjust to stimulus onset within less than a second, their relaxation back to resting state after stimulus stops is longer in E-E connectivity (about 6s) than in E-I connectivity (less than a second). This longer delay in E-E connectivity is due to the positive feedback gain between the two areas: if we focus for instance on Control area, even after the stimulus has stopped, there is still some overactivation (w.r.t. basic forcing) due to stimulation by Target area, which in turn is sustained by the remaining activity level of Control area (and vice versa). Here, the positive feedback gain acts as a brake on relaxation.

By contrast, in E-I connectivity, the high activity level of Control area tends to decrease Target activity (w.r.t. resting state) so that there is an under-activation of Control area (even if less and less as relaxation proceeds) so that nothing slows down the return to resting state (a negative feedback gain would not act as a brake).

As a consequence of its long characteristic time, the BOLD signal does not have time enough to reach its steady state value in the case of a series of short stimulations: it oscillates around values slightly below the steady state value observed for the long stimulation. In such stimulation series pattern, the time-shifted bijectivity between synaptic activity and BOLD signal is lost.

Still, considering the short time adjustment of synaptic activities levels to stimulus onset, it remains relevant to consider finite differences between stationnary synaptic activities at stimulated state vs. non stimulated state as a valuable observable of system response once all feedback loops are taken into account. For example, in [Fig 12], $\Delta sn_C^* = (0.8 - 0.2) = 0.6$ and $\Delta sn_T^* = (0.6 - 0.2) = 0.4$ are a good identification of the system's response to a stimulus of $\Delta B_{E_C} = 0.1$ nA in Control area in an E-E connectivity.

**3.4.2 Effect of self-inhibitory strength within Target area depending on connectivity.** To understand how parameters can drive system dynamics through such feedback loops, we chose to look at the cross-effect between the self-inhibitory parameter $J_{gaba_T}$ of the Target area and the ability of the Control area to have an effect upon the Target area, when submitted to a step activation $\Delta B_{E_C} \equiv I_{stim}$.

As for the range explored for $J_{gaba_T}$, we set it with regards to the values that would ensure resting states at 3 Hz in both areas, from the minimal value for EI and II connectivities (0.929 nA) up to about twice the value for EE and IE connectivities (1.758 nA). The response $sn_C$ and $sn_T$ are presented in [Fig 13] for the four connectivities (BOLD signal would display corresponding changes).

To make the link with previous sections, we outline, as a first point, that the response of $sn_C$ and $sn_T$ to a finite step of activation of Control area can be read as a continuous sum over $I_{stim}$ of the sensitivities that we have established analytically in previous section. We can then understand that, for any value of $J_{gaba_T}$, integrating always positive or always negative sensitivities, the sign depending on the kind of connectivity, will translate into the same sign of response of Target area: E-E and E-I connectivities will trigger Target up-regulation, and only I-E and I-I connectivities can account for inhibitory control, where Target activity is down-regulated by activation of Control activity.

Focusing now on responses with $J_{gaba_T}$ set to values of reference (those ensuring 3 Hz in both areas at rest, dotted lines in [Fig 13]), we observe well the non linear effect of stimulation upon responses in both area, resulting from the integration of sensitivities. For instance, in E-E connectivity, the sharp increase in $sn_C$ and $sn_T$ for a slight value of $I_{stim} \simeq 0.02$ nA results for the integration of the sharp curves of sensitivities in [Figs 8C] and [8D] just beyond $x_{E,\text{ref}}$ and for $J_{gaba_T} = 1.75$ nA. As a consequence, further steps of activation can not but have a lower and lower effect on target area $sn_T$, and the range for control is limited. It is also the case for I-I connectivity. As anticipated from sensitivities, responses in I-E and E-I connectivities are more gradual.

If we now consider affecting $J_{gaba_T}$ while keeping $J_{gaba_C}$ unaffected (namely set at the value of reference w.r.t. 3 Hz resting state), the effect of feedback loop from Target to Control can be significant upon the activity of Control area, even at resting state. It can even lead to unrealistic values for resting firing rates. For instance in EE connectivity, for $J_{gaba_T} = 1.0$ nA (instead of 1.758), we get $sn_C = 0.58$ and $sn_T = 0.74$, corresponding respectively to $rn_C = 15.7$ Hz and $rn_T = 32.0$ Hz.

In the same way, upon stimulation, applying the same step activation (e.g., $I_{stim} = 0.02$ nA) not only will affect differently $sn_T$ depending on $J_{gaba_T}$, but will also affect the corresponding $sn_C$ in Control Area, as a result of feedback loops.

As a consequence, since the control effectivity is to be understood as the contrast between resting values and activated values, a major point is that changing $J_{gaba_T}$ not only affects stimulated states but also affects resting values. Hence, $J_{gaba_T}$ affects control effectivity through two factors: activity levels before stimulation is applied, and the margin for increase or decrease.

Overall, all these results show the major role of the within-area self-inhibition in Target area, depending on the connectivity.

# 4 Conclusion

In this study, our aim was to understand how brain regions can be connected at neural level so that, in a Target area, a decrease in BOLD activity would be observed when a Control area is subjected to an increase in BOLD activity. For the neural model, we took inspiration from the model by Naskar et al. [1], which describes the mean synaptic activities, both excitatory and inhibitory within each area. More specifically, we used the same differential equations and set of

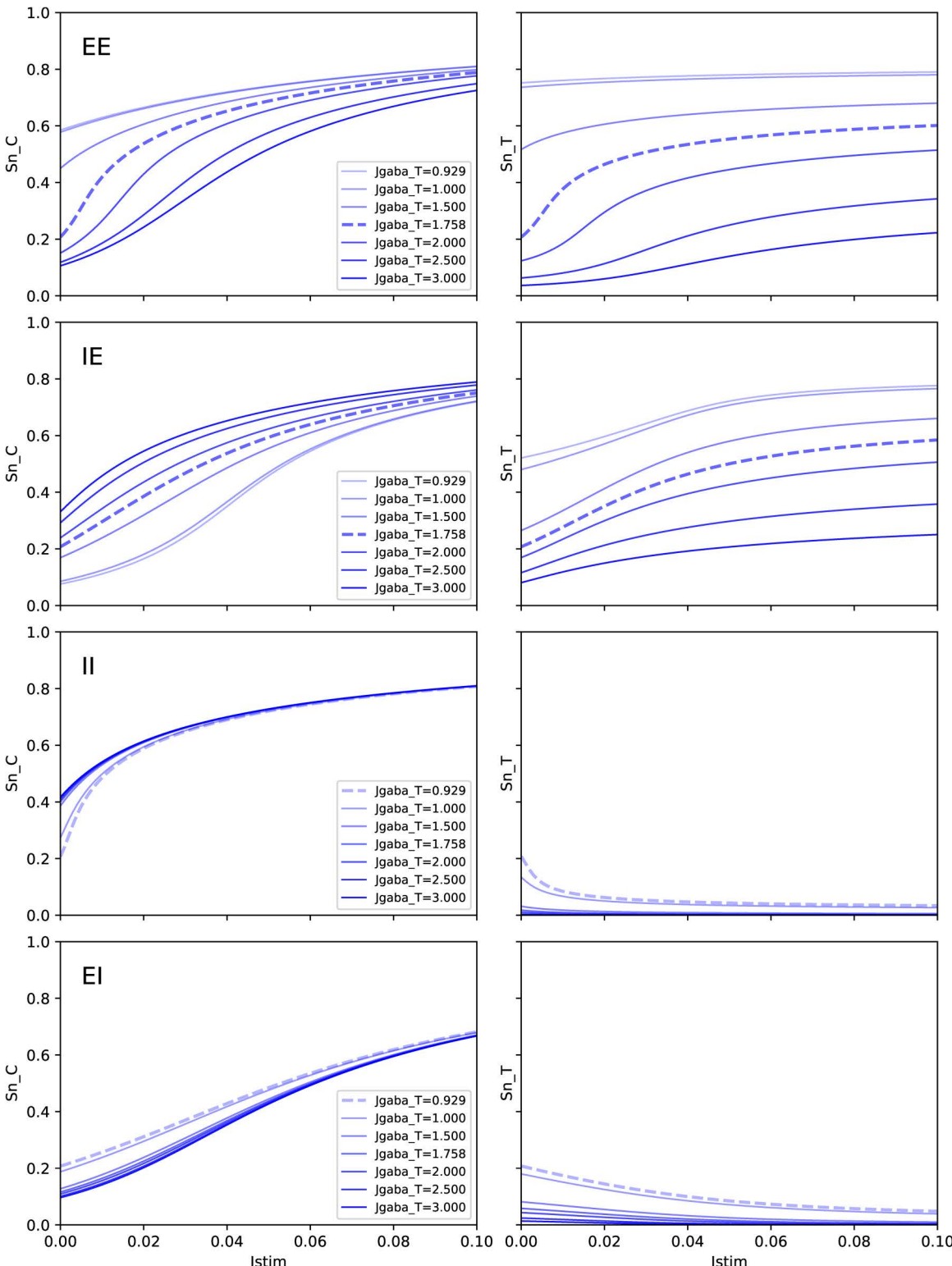

**Fig 13. Effect of varying $J_{gaba_T}$, the self-inhibitory strength within Target area, upon response to step activation of Control area.** In order by lines: EE, IE, II, EI. Left: Control area, Right: Target area. Dotted lines correspond to the value of $J_{gaba_T}$ yielding firing rates at 3 Hz in both areas when the system is at rest ($I_{stim} = 0$). See comments in Sec 3.4.2.

neurobiological parameters, including their proposed range of values. Their model includes an implicit type of connectivity which we termed E-E. For the coupling from neural state to BOLD signal, we used the Extended Balloon-Windkessel model as proposed by [33]. When this composite model (neural and BOLD) was studied as a dynamical model, it was observed to reach, quite rapidly, a stationary neural state (also called a fixed point, FP), as opposed to the BOLD signal which follows the fluctuations in neural signal with a certain delay and would hardly reach a FP. We have therefore focused on this neural model in its stationary state, as a first step. Using this as a foundation, we built a sensitivity analysis of the FP using two brain areas as a configuration model, more specifically, a control area and a target area. Using this two-areas model, we built a hierarchical system of nested sensitivities and we have shown that:

- To have a decrease in synaptic activity in a Target area, we needed to consider other types of connectivity than E-E, as it is unable to cause an inhibitory response. Four different types of connectivities have been explored, and we show that only E-I and I-I connectivities can reduce the synaptic activity in the Target region.

- The E-I and I-I connectivities, which both enable inhibitory control, contrast with each other in terms of the correlation between changes in activity in control and target areas, with possibly E-I allowing a smoother control.

- The intra-area self-inhibitory coefficient $J_{gaba_T}$ in the Target area modulates this decrease in synaptic activity when the Control region is stimulated. By analyzing the response to a finite perturbation of the external forcing in the Control area as a function of $J_{gaba_T}$ in the Target area, we observed that $J_{gaba_T}$ affects both resting and activated states.

## 5  Perspectives

With regards to our methodology to build analytical expression for sensitivity by nesting up across levels of description, there is actually no difference, in terms of validity perimeter and numerical concerns, between our nesting methodology and a brute-force formalism that would get as input the complete system and derive the Jacobian-based sensitivities. As a formal proof, and because any ambiguity on this point must be discarded, we provide an additional SI which exposes the convergence of the two methods (S4 File).

By contrast, in terms of legibility, physical meaning, and practicality, the two methods differ completely. For instance, as illustrated in the development for a 3-Areas system (S5 File), one can see that expressing sensitivities incorporating an additional area in the system does not require to start from scratch the derivation of the Jacobian-based sensitivity for the full system with 3-areas, but can be built more intelligibly by leveraging the nested approach: expressions for open-loop area sensitivities are the same, so one has just to incorporate one and one only further element: the new inter-areas connectivity. This also applies to any number N regions.

As for our perspectives, we then currently work on the generalization of the method for building nested sensitivities for more complex network configurations, e.g., a system with several control and target areas grouped into two sub-systems: a sub-system of Control areas and a sub-system of Target areas. Such an approach would yield an *analytical* access to sensitivities for some configurations proposed in the context of inhibitory control, such as the one by Mary et al. [32] which points out a network of 14 regions, comprising of 9 Control and 5 Target areas as a drive for the inhibitory control of intrusive memory in Post-Traumatic Stress Disorder. Furthermore, this generalization to a system-level with more areas would allow for the integration of relays in the network, as observed in recent studies [34]. Analytical sensitivities could also help identifying areas that mostly influence information flow in any brain network, with no need to fall back to numerical experiments [35,36].

## Supporting information

**S1 File. Parameters values of reference as found in Naskar et al. [1].**
(PDF)

**S2 File. Formal developments for sensitivities and linear stability analysis.**
(PDF)

**S3 File. Python codes to generate all figures.**
(ZIP)

**S4 File. Jacobian-based derivations.**
(PDF)

**S5 File. Three coupled area system.**
(PDF)

## Author contributions

**Conceptualization:** Anaïs Vallet, Francis Eustache, Jacques Gautrais, Pierre Gagnepain.

**Formal analysis:** Anaïs Vallet, Stéphane Blanco, Coline Chevallier, Jacques Gautrais, Jean-Yves Grandpeix, Jean-Louis Joly, Shailendra Segobin.

**Funding acquisition:** Francis Eustache, Pierre Gagnepain.

**Methodology:** Anaïs Vallet, Stéphane Blanco, Coline Chevallier, Jean-Yves Grandpeix, Jean-Louis Joly, Shailendra Segobin.

**Project administration:** Francis Eustache, Pierre Gagnepain.

**Software:** Anaïs Vallet, Jean-Louis Joly.

**Supervision:** Francis Eustache, Jacques Gautrais, Pierre Gagnepain.

**Validation:** Francis Eustache, Jacques Gautrais, Pierre Gagnepain.

**Visualization:** Anaïs Vallet, Jean-Louis Joly.

**Writing – original draft:** Anaïs Vallet, Jacques Gautrais, Jean-Yves Grandpeix, Jean-Louis Joly, Shailendra Segobin, Pierre Gagnepain.

**Writing – review & editing:** Anaïs Vallet, Francis Eustache, Jacques Gautrais, Jean-Yves Grandpeix, Jean-Louis Joly, Shailendra Segobin, Pierre Gagnepain.

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
