## [Decision Letter · Decision Letter 0]

6 Aug 2025

Sensitivity analysis enlightens effects of connectivity in a Neural Mass Model under Control-Target mode

PLOS Computational Biology

Dear Dr. Gautrais,

Thank you for submitting your manuscript to PLOS Computational Biology. After careful consideration, we feel that it has merit but does not fully meet PLOS Computational Biology's publication criteria as it currently stands. Therefore, we invite you to submit a revised version of the manuscript that addresses the points raised during the review process.

Please submit your revised manuscript within 60 days Oct 06 2025 11:59PM. If you will need more time than this to complete your revisions, please reply to this message or contact the journal office at ploscompbiol@plos.org. Please include the following items when submitting your revised manuscript:

We look forward to receiving your revised manuscript.

Kind regards,

Arvind Kumar, Ph.D.

Academic Editor

PLOS Computational Biology

Hugues Berry

Section Editor

PLOS Computational Biology

**Additional Editor Comments:**

Your paper was reviewed by two experts. They both liked the paper but they have made several suggestions. So I would like to invite you consider those and revise your manuscript accordingly.

**Journal Requirements:**

At this stage, the following Authors/Authors require contributions: Jacques Gautrais. Please ensure that the full contributions of each author are acknowledged in the "Add/Edit/Remove Authors" section of our submission form.

4) Your manuscript is missing the following sections: Results, and Methods.  Please ensure all required sections are present and in the correct order. Make sure section heading levels are clearly indicated in the manuscript text, and limit sub-sections to 3 heading levels. An outline of the required sections can be consulted in our submission guidelines here:

5) Please upload all main figures as separate Figure files in .tif or .eps format. For more information about how to convert and format your figure files please see our guidelines:

2) If any authors received a salary from any of your funders, please state which authors and which funders..

7) Please ensure that the funders and grant numbers match between the Financial Disclosure field and the Funding Information tab in your submission form. Note that the funders must be provided in the same order in both places as well. Currently, the Financial Disclosure states there was no funding received.

**Reviewers' comments:**

Reviewer's Responses to Questions

**Comments to the Authors:**

Reviewer #1: Thank you of this paper. It discusses a technical but important aspect of understanding the excitatory inhibitory balance of the brain. The analysis is sound and correct. The derivations are easy to follow. There a few comments on the paper.

Major comments:

The paper studied how the stationary points of a dynamic system with single and collections of neuronal pools change depending on the internal connectivity and external forcing. The stability of a stationary point was partly explained for the single state system (excitation or inhibition only) but I did not see a stability analysis for the interconnected neural pools. This paper would gain considerably with an analytical section on the stability of the stationary points for each different connectivity set-up.

Minor comments:

1) The notation is not useful for a reader not accustomed to the specific references noted in the paper. The use of e.g. sn and sg for the state of the NMDA or GABA looks more like a product between s and n or g. It would be useful to use a specific notation for the terms (greek letters or other specific math symbols).

2) In equation 1 is $\sigma v_i$ a random noise? and if so how was it dealt with in deriving equation 5. Are there underlying assumptions of stability of the system? Again a stability analysis would be useful.

3) Line 185: What is canonical with equation 10. It is a useful expression, and used repeatedly in the manuscript but I do not see why it is called canonical.

4) Line 188: What does this mean: ...non linearity is the self-amplification of the excitatory pool by the recurrent connectivity, which is expressed in the function wn through the parameter W+.

5) line 207: The idea of connection gains having values above and below 0 is a new interesting constraint on the dynamics. This was partly discussed where the connection gain was treated as a gauge field in Cooray GK, Cooray V, Friston KJ. Cortical dynamics of neural-connectivity fields. Journal of Computational Neuroscience. 2025 Apr 10:1-9.

Reviewer #2: The paper presents an interesting approach to include and study the effects of inhibitory connectivity in phenomemological models such as neural field/mass models. The authors use the framework of sensitivity analysis to see how perturbation in one area (control) affects the activvation in another area (target), where the perturbed parameter is related to external input currents. The equations are modified to allow for both long-range excitatory and inhibitory connections (eq.4), which esssentially includes modulating the fraction of long range excitatory connections and rest is projected to the inhibitory population. Overall, a mathematically thorough and well-written paper.

While the idead is interesting and probably also has much need to be explored for development of modeling work in neuroscience, there are few major concerns that need to be addressed before I am convinced.

1. I am not very convinced with the idea of using just two-regions to demonstrate the concept. In such scenarios, the simplest case that should be tested is a three region model. May some examples of a structured sub-network involving three regions could be used (for ex. something from frontostriatal circuit ?)

2. Relating the change in sign of the derivative to iup or down-regulation is straightforward and trivial in a two region scenario. Again, the iossue is that when it is scaled to three or more regions is this interpretation valid ? As the effect could pass through many (intermediate) areas ? This should be discussed and the validity beyond two-region scenario should be throroughly discussed.

3. The nested sensitivity approach taken by the authors assume separability (if I am not mistaken). What would be the author's view on using Jacobian matrix instead ? Probably, a comparison with Jacobian-based sensitivityy would clarify the validity or limits of nestd sensitivity?

4. Regarding the structure of the paper, it would help to have a brief into to sensitivity analysis and its applications to neural mass models as part of lit review (ex. see https://doi.org/10.1103/PhysRevE.110.044208 ) as well as other approaches to understand parameter importance (PLoS Comput. Biol. 14, e1006009 (2018), PLoS Comput. Biol. 15, e1006694 (2019)).

**Have the authors made all data and (if applicable) computational code underlying the findings in their manuscript fully available?**

Reviewer #1: Yes

Reviewer #2: **No:** i could not find any git repo linked in the paper

PLOS authors have the option to publish the peer review history of their article (what does this mean? ). If published, this will include your full peer review and any attached files.

**Do you want your identity to be public for this peer review?** For information about this choice, including consent withdrawal, please see our Privacy Policy .

Reviewer #1: **Yes:** Gerald Cooray

Reviewer #2: No

**Figure resubmission:**

**Reproducibility:**



---

## [Decision Letter · Decision Letter 1]

23 Dec 2025

PCOMPBIOL-D-25-00851R1

Sensitivity analysis enlightens effects of connectivity in a Neural Mass Model under Control-Target mode

PLOS Computational Biology

Dear Dr. Gautrais,

Thank you for submitting your manuscript to PLOS Computational Biology. After careful consideration, we feel that it has merit but does not fully meet PLOS Computational Biology's publication criteria as it currently stands. Therefore, we invite you to submit a revised version of the manuscript that addresses the points raised during the review process.

We look forward to receiving your revised manuscript.

Kind regards,

Arvind Kumar, Ph.D.

Academic Editor

PLOS Computational Biology

Hugues Berry

Section Editor

PLOS Computational Biology

**Additional Editor Comments:**

The two reviewers are now happy with the revision of the manuscript. There is one outstanding issue: the reviewers would like to have a look at the code. I read your note in the rebuttal letter. Until the reviewers are fully satisfied I cannot promise a positive outcome but since you have addressed all the concerns raised the only remaining issue is the code. As you may have seen by now it has become a norm and not just PloS Comp most journals would require you to submit the code to reproduce key results.

**Journal Requirements:**

1) We have noticed that you have uploaded Supporting Information files, but you have not included a list of legends. Please add a full list of legends for your Supporting Information files after the references list.

**Reviewers' comments:**

Reviewer's Responses to Questions

**Comments to the Authors:**

Reviewer #1: No further comments, thanks you.

Reviewer #2: I am satisfied with the responses. However, I cannot find any code/Data that is made available as per journal policies. Kindly make this available and then I happy to accept.

**Have the authors made all data and (if applicable) computational code underlying the findings in their manuscript fully available?**

Reviewer #1: Yes

Reviewer #2: **No:**

PLOS authors have the option to publish the peer review history of their article (what does this mean? ). If published, this will include your full peer review and any attached files.

**Do you want your identity to be public for this peer review?** For information about this choice, including consent withdrawal, please see our Privacy Policy .

Reviewer #1: **Yes:** Gerald K. Cooray

Reviewer #2: No

**Figure resubmission:**
---

## [Decision Letter · Decision Letter 2]

17 Feb 2026

Dear Dr. Gautrais,

We are pleased to inform you that your manuscript 'Sensitivity analysis enlightens effects of connectivity in a Neural Mass Model under Control-Target mode' has been provisionally accepted for publication in PLOS Computational Biology.

Best regards,

Arvind Kumar, Ph.D.

Academic Editor

PLOS Computational Biology

Hugues Berry

Section Editor

PLOS Computational Biology

Thank you for your patience with the review process. I am pleased to inform that now we can accept your paper in its current form. Congratulations for a very fine contribution.

Reviewer's Responses to Questions

**Comments to the Authors:**

Reviewer #2: Thanks for providing the codes.

**Have the authors made all data and (if applicable) computational code underlying the findings in their manuscript fully available?**

Reviewer #2: Yes

PLOS authors have the option to publish the peer review history of their article (what does this mean? ). If published, this will include your full peer review and any attached files.

**Do you want your identity to be public for this peer review?** For information about this choice, including consent withdrawal, please see our Privacy Policy .

Reviewer #2: **Yes:** Narayan Puthanmadam Subramaniyam

---

## [Editor Report · Acceptance letter]

PCOMPBIOL-D-25-00851R2

Sensitivity analysis enlightens effects of connectivity in a Neural Mass Model under Control-Target mode

Dear Dr Gautrais,

I am pleased to inform you that your manuscript has been formally accepted for publication in PLOS Computational Biology. Your manuscript is now with our production department and you will be notified of the publication date in due course.

With kind regards,

Judit Kozma
